# Multi-satellite sensor study on precipitation-induced emission pulses of $NO_x$ from soils in semi-arid ecosystems

J. Zörner[1], M.J.M. Penning de Vries[1], S. Beirle[1], H. Sihler[1,3], P.R. Veres[2,*], J. Williams[2], and T. Wagner[1]

[1]Satellite Remote Sensing Group, Max Planck Institute for Chemistry, Mainz, Germany
[2]Atmospheric Chemistry Department, Max Planck Institute for Chemistry, Mainz, Germany
[3]Institute of Environmental Physics, University of Heidelberg, Heidelberg, Germany
[*]now at: NOAA Earth System Research Laboratory, Boulder, USA

*Correspondence to:* Jan Zörner (jan.zoerner@mpic.de)

**Abstract.** We present a top-down approach to infer and quantify rain-induced emission pulses of $NO_x$ ($\equiv NO + NO_2$), stemming from biotic emissions of NO from soils, from satellite-borne measurements of $NO_2$. This is achieved by synchronizing time series at single grid pixels according to the first day of rain after a dry spell of prescribed duration. The full track of the temporal evolution several weeks before and after a rain pulse is retained with daily resolution. These are needed for a sophis-

ticated background correction, which accounts for seasonal variations in the time series and allows for improved quantification of rain-induced soil emissions. The method is applied globally and provides constraints on pulsed soil emissions of $NO_x$ in regions where the $NO_x$ budget is seasonally dominated by soil emissions.

We find strong peaks of enhanced $NO_2$ Vertical Column Densities (VCDs) induced by the first intense precipitation after prolonged droughts in many semi-arid regions of the world, in particular in the Sahel. Detailed investigations show that the

rain-induced $NO_2$ pulse detected by the OMI, GOME-2 and SCIAMACHY satellite instruments could not be explained by other sources, such as biomass burning or lightning, or by retrieval artefacts (*e.g.* due to clouds).

For the Sahel region, absolute enhancements of the $NO_2$ VCDs on the first day of rain based on OMI measurements 2007–2010 are on average $4 \times 10^{14}$ molec cm$^{-2}$ and exceed $1 \times 10^{15}$ molec cm$^{-2}$ for individual grid cells. Assuming a $NO_x$ lifetime of 4 hours, this corresponds to soil $NO_x$ emissions in the range of 6 ng N m$^{-2}$ s$^{-1}$ up to 65 ng N m$^{-2}$ s$^{-1}$, in good agree-

ment with literature values. Apart from the clear first-day peak, $NO_2$ VCDs are moderately enhanced ($2 \times 10^{14}$ molec cm$^{-2}$) compared to background over the following two weeks suggesting potential further emissions during that period of about 3.3 ng N m$^{-2}$ s$^{-1}$. The pulsed emissions contribute about 21-44% to total soil $NO_x$ emissions over the Sahel.

# 1 Introduction

Nitrogen oxides ($NO_x \equiv NO + NO_2$) play an important role in tropospheric chemistry. They are key catalysts in chemical processes generating and destroying ozone ($O_3$) (Crutzen, 1995; Crutzen and Lelieveld, 2001; Seinfeld and Pandis, 2006). Ambient mixing ratios of $NO_x$ and volatile organic compounds (VOCs) determine whether tropospheric $O_3$ is formed or consumed (Chameides et al., 1992; Crutzen and Lelieveld, 2001). In polluted conditions with high $NO_x$, through the reaction of nitric oxide (NO) with the hydroperoxyl radical ($HO_2$), $NO_x$ is also involved in the production of the hydroxyl radical (OH) and, thus, impacts the tropospheric oxidizing capacity (Monks, 2005). During daytime, $NO_x$ is mainly removed from the atmosphere by oxidation with OH producing nitric acid ($HNO_3$) (Jacob, 1999; Monks, 2005) which is an important component of acid deposition and contributes to nitrate aerosol formation (Bassett and Seinfeld, 1983). Furthermore, $NO_x$ is also removed by $NO_2$ deposition on vegetated surfaces (Ganzeveld et al., 2002). At night, $N_2O_5$ hydrolysis on aerosol surfaces is the dominant sink of $NO_x$ (Jacob, 1999).

While anthropogenic activity such as fossil-fuel combustion is the largest source of $NO_x$, there are also important natural sources including natural biomass burning from forest fires, lightning and microbial processes in soils. Soil emissions of $NO_x$ ($sNO_x$) constitute an estimated fraction of $\approx 15\%$ of total $NO_x$ on a global basis (Warneck and Williams, 2012; Hudman et al., 2012) and may even dominate the local $NO_x$ budget in non-industrialized regions like remote tropical and agricultural areas (Yienger and Levy, 1995; Steinkamp et al., 2009). Bottom-up approaches using global chemistry models suggest global fluxes between 4 and 15 $Tg\,N\,yr^{-1}$ with uncertainties of up to 5–10 $Tg\,N\,yr^{-1}$ (e.g., Yienger and Levy, 1995; Steinkamp and Lawrence, 2011; Hudman et al., 2012; Vinken et al., 2014, and references therein). Satellite constrained top-down approaches hint at regional underestimations of $sNO_x$ of a factor of 2 and more (Jaeglé et al., 2004; Wang et al., 2007; Boersma et al., 2008; Zhao and Wang, 2009; Vinken et al., 2014). Thus, global emissions of $sNO_x$ remain uncertain.

Emissions of $NO_x$ from natural and anthropogenically influenced soils are mainly driven by microbial activity within the top soil layer and associated chemical reactions (Conrad, 1996). Primarily, two important groups of micro-organisms, nitrifiers and denitrifiers, are involved in processes related to the turnover of nutrients in the soil (Pilegaard, 2013; Behrendt et al., 2014). They are directly responsible for the corresponding processes of: (i) nitrification, the biological oxidation of nitrogen compounds, typically the oxidation of soil ammonium ($NH_4^+$) to nitrate ($NO_3^-$) and (ii) denitrification, the reduction of nitrate by microbes to gaseous products, i.e. $N_2O$ and finally $N_2$. NO is a gaseous by-product of both processes and once released reacts with ambient $O_3$, to form $NO_2$ and oxygen ($O_2$) within minutes. Findings from Oswald et al. (2013) suggest that gaseous nitrous acid (HONO), which is rapidly photolyzed to NO, is also emitted from soils.

Most soil emissions of NO in semi-arid areas are linked to microbial processes, but some chemical (abiotic) formation processes of NO are known to exist, which are more important for acidic soils with high nitrite ($NO_2^-$) concentrations (Davidson, 1992b). Such soils are found in humid regions, i.e. the humid tropical belt and the northern temperate zone, where soil leaching removes alkaline material and associated salts from the soil profiles leading to pH-values of less than 5.5 (Merry, 2009). Emissions of nitrogen-containing gases, such as NO, $N_2$ and $N_2O$ increase dramatically in soils with enhanced nitrogen availability due to the presence of N-fixing microbial species and plants (Virginia et al., 1982; van Groenigen et al.,

2015). In semi-arid areas with sparse vegetation cover, surfaces are covered by a variety of communities of cyanobacteria, algae, lichens, mosses, microfungi, and other bacteria in differing proportions (Belnap and Lange, 2001; Barger et al., 2005). Associated organisms within these biological crusted soils fix atmospheric $N_2$ and, thus, raise the nitrogen availability in the soil (Evans and Ehleringer, 1993). Barger et al. (2005) found varying rates of soil NO fluxes from biologically crusted soils

that differed in their nitrogen fixation potentials. Recent findings from Weber et al. (2015) suggest that dryland emissions of reactive nitrogen are largely driven by biocrusts rather than the underlying soil and strongly depend on the soil water content (SWC), i.e. precipitation events. Throughout this study, for simplicity all emissions from soils and biocrusts are referred to as $sNO_x$.

Soil emissions of trace gases depend on a wide range of ambient environmental conditions such as soil type, soil moisture,

temperature, pH-Value and nitrogen content (Conrad, 1996; Ludwig et al., 2001; Meixner and Yang, 2006; Oswald et al., 2013). Also agricultural management practices such as soil cultivation, fertilization and irrigation can strongly affect the fluxes (Bouwman et al., 2002). In remote regions like the Sahel, where synthetic fertilizer is limited, manure plays a prominent role in the fertilization of agricultural fields and can contribute significantly to the input of organic nitrogen into the soil (Schlecht and Hiernaux, 2004; Delon et al., 2010). The effective $NO_x$ fluxes from soils to the atmosphere are potentially offset by "canopy

reduction" where nitrogen oxides are quickly deposited on available vegetation surfaces (Ganzeveld et al., 2002). During the dry season in tropical ecosystems soils accumulate inorganic nitrogen through N-fixing micro-organisms. Subsequently, water-stressed microbes trapped in the soil become activated by the first rain event of the wet season and release NO as a by-product of nitrogen consumption (Davidson, 1992a). Rain-induced pulsing events of $NO_x$ emissions were observed in-situ and by laboratory measurements of soil samples (e.g., Williams et al., 1987; Johansson and Sanhueza, 1988; Williams et al., 1992;

Levine et al., 1996; Scholes et al., 1997; Kim et al., 2012; Wang et al., 2015). Pulsed emissions of $sNO_x$ occurring at the transition phase between the dry and wet season, were previously also observed from space (Jaeglé et al., 2004; Bertram et al., 2005; Hudman et al., 2012) in the Sahel region. Hudman et al. (2012) showed that intense but short events of soil emissions, i.e. pulsed emissions, at the start of the wet season after a prolonged dry spell represent a large fraction of annual soil emissions in the Sahel region. As noted by Hudman et al. (2012) further research needs to be done to verify that the observed pulses by

OMI (Ozone Monitoring Instrument) are not biased by the retrieval algorithm.

The main objective of this study is to quantify precipitation-induced short-term enhancements in soil emissions of $NO_x$, which show peak emissions on the scale of 1-3 days, from space-based instruments in semi-arid regions in the world. This is achieved by investigating the evolution of tropospheric $NO_2$ column densities from multiple satellite sensors before and after the first rainfall events on the onset of the wet season.

We introduce an optimized algorithm that synchronizes and averages multiple time series of atmospheric variables either from one location only, or from individual grid pixels, by aligning them on a relative time scale to each other. Our algorithm enhances the basic approach described by Hudman et al. (2012) with several features: (i) performing the analysis globally with (ii) high spatial resolution, which is both achieved by expanding the time span of the study to several years (2007 to 2010) enabling an investigation of single grid pixels of $0.25°$ with reasonable statistics. (iii) The full track of the temporal evolution

several weeks before and after a rain pulse is retained with daily resolution. (iv) By intercomparing measurements of $NO_2$ from

multiple satellite instruments it is possible to quantify potential measurement artefacts and investigate the impact of different retrieval algorithms. Furthermore, sensitivity studies are conducted in order to evaluate the impact of the a-priori assumptions on thresholds for daily rainfall, i.e. the definition of drought, and its requested duration.

Our approach is a purely *top-down* method, in the sense that satellite data of trace gases are exclusively used to describe and quantify phenomena taking place on the Earth's surface and atmosphere. It is, therefore, extremely important to consider natural processes in the atmosphere that could trigger soil emissions or may affect the retrieved $NO_2$ column densities in other ways. In order to achieve this we incorporate total columns of water vapour, humidity, temperature and wind directions in our analysis to assess the prevailing meteorology. To verify that the observed responses in the trace gas column densities reflect the impact of emissions fluxes from the soil, possible interferences from other parameters, *e.g.* fires, modified cloud fractions, coincidences with lightning events and horizontal transport from polluted regions, are also investigated.

Veres et al. (2014) found in laboratory experiments that also several volatile organic compounds (VOC) including HCHO exhibit pulsed emissions when dry soils are first wetted. Our study, hence, also addresses the question whether HCHO emissions from semi-arid soils can be observed from satellite-borne sensors.

In contrast to previous satellite studies, our study makes a clear distinction between (i) pulsed emissions which show strong gradients on a day-to-day scale triggered by a singular precipitation event and (ii) background emissions which are not directly affected or could not be unambiguously related to a strong precipitation pulse. This facilitates the assessment of the contribution from single $sNO_x$ pulses additionally to background levels.

The paper is organized as follows: in section 2, all data products used within this study are presented. In section 3, the basic algorithm used for averaging the time series of environmental parameters along a relative time axis around the first day of precipitation is described. In section 4, this approach is then applied to areas with different spatial extents. We first perform an analysis on a global scale to delineate regions that show pronounced features in $sNO_x$ in response to the first rain after a prolonged dry spell. In a second step, we focus on Africa and the Sahel region, in specific, and separate the analysis for different seasons. For this region, we investigate fundamental relationships between soil emissions and some of their governing parameters, i.e. soil moisture content, temperature, air humidity. Within this analysis possible interferences from other parameters are also investigated, and detailed sensitivity studies are conducted. In section 5, $sNO_x$ emissions are inferred from the $NO_2$ VCDs based on a sophisticated background correction.

## 2 Data

### 2.1 Satellite observations of trace gases

Vertical column densities (VCDs) of $NO_2$ and HCHO can be retrieved from nadir-viewing satellite instruments, by analysing solar backscatter radiances in the UV-VIS spectral range. Differential Optical Absorption Spectroscopy (DOAS, Platt and Stutz, 2008), which exploits characteristic narrow absorption structures, is typically used for the analysis.

Tropospheric VCDs are usually derived in a multi step process (e.g. Boersma et al., 2004, 2007; De Smedt et al., 2008, 2012). First, total slant column densities (SCDs) are retrieved, i.e. the integrated concentrations along the effective light path, by fitting

the measured spectrum with a model taking into account all other absorbers in the atmosphere. Second, tropospheric SCDs are derived by subtracting the stratospheric column ($NO_2$) or a latitude-dependent bias estimated over the Pacific (HCHO). Third, the tropospheric SCDs are then translated to tropospheric VCDs. The conversion of SCDs to VCDs is usually performed by dividing the SCDs by a so called air mass factor (AMF) (Solomon et al., 1987). The AMF is derived from radiative transfer simulations taking into account information of ground albedo, aerosols and clouds, the vertical profile of the trace gas and the satellite viewing geometry (e.g. Palmer et al., 2001; Richter and Burrows, 2002; Martin, 2003).

Except for very low tropospheric trace gas amounts, the tropospheric AMF dominates the uncertainty of tropospheric trace gas observations from space, *e.g.* caused by insufficient knowledge of the trace gas profiles, aerosols, cloud properties and cloud cover (Boersma et al., 2004). For observations of trace gases in the boundary layer, which are the focus of this study, the main effect of clouds is that they shield the atmosphere below. Thus, in the presence of clouds the retrieved trace gas absorptions (SCDs) are usually decreased compared to clear sky conditions. During the conversion to VCDs, the AMF compensates for this effect which, however, might lead to over- or underestimations of the trace gas column densities if the state of the atmosphere is not known precisely. As the transition between days with and without precipitation generally corresponds to a change in cloud cover, cloud effects need to be investigated. Therefore, satellite measurements retrieved under low and high cloud fractions are studied in detail using cloud information derived from FRESCO+ (Wang et al., 2008) for GOME-2 and SCIAMACHY and OMCLDO2 (Acarreta et al., 2004) for OMI.

## 2.2 Satellite instruments and trace gas products

SCIAMACHY (Bovensmann et al., 1999) aboard the ENVISAT satellite was operated from 2002 to 2012. It had a ground pixel size of about 30 x 60 $km^2$ (vis) to 30 x 120 $km^2$ (UV). GOME-2 (Callies et al., 2000; Munro et al., 2006) aboard ESA's METOP-A satellite, launched in 2007, has a ground pixel size of about 40 x 80 $km^2$. OMI (Levelt et al., 2000, 2006) on NASA's Aura platform, which was launched in 2004 has a ground pixel size of 13 x 24 $km^2$ at nadir and increasing pixel sizes to the far ends of the 2600 $km$ wide swath. In our study, the two outermost pixels are screened out to remove the pixels with the largest viewing angles and lowest spatial resolution. The local overpass times for the three satellite instruments at the equator are about 9:30 a.m. for GOME-2, 10:00 a.m. for SCIAMACHY and 1:30 p.m. for OMI.

For $NO_2$, the products GOME-2 TM4-NO2A version 2.3, SCIAMACHY TM4-NO2A version 2.3 and OMI DOMINO version 2.0.1 are used (Boersma et al., 2004, 2011). For HCHO the products GOME-2 version 12, SCIAMACHY version 12 and OMI version 14 are used (De Smedt et al., 2012). Data products are provided freely by the Tropospheric Emission Monitoring Internet Service (TEMIS) via http://www.temis.nl/.

Differences among the trace gas data products from the three satellite instruments are expected due to, among others, the calculation of the AMF, their different ground pixel size, local overpass time, cloud products used, the diurnal cycle of cloud conditions and the covered time period. Furthermore, the diurnal cycle of the instantaneous $NO_x$ lifetime and emissions might also cause systematic differences between SCIAMACHY and GOME-2 on the one hand, and OMI on the other.

Uncertainties of tropospheric $NO_2$ VCDs result mainly from uncertainties of the stratospheric correction (about $2 * 10^{14}$ molec $cm^{-2}$) and tropospheric AMFs (about 35-60%) (Boersma et al., 2004).

## 2.3 Precipitation

The estimation of precipitation on a daily global scale is facilitated through the combination of radar, passive microwave, visible (VIS) and infrared (IR) sensors aboard low-earth orbiting as well as geostationary satellites. In this study, three different products are used which employ such a blended precipitation scheme. All three data sets agree in their spatial resolution (0.25° x 0.25°) and provide data in 3-hourly time steps, i.e. 12UTC covering the period 22:30UTC to 1:30UTC and so on. They are briefly described below.

The Tropical Rainfall Measuring Mission (TRMM) Multisatellite Precipitation Analysis (TMPA) 3B42 Version 7 dataset (Huffman et al., 2007) combines observations made by the TRMM satellite with other satellite systems, as well as land surface precipitation gauge analyses when possible. The passive microwave data, which has a strong physical relationship to the hydrometeors that result in surface precipitation, are collected from the Microwave Imager (TMI) on TRMM, Special Sensor Microwave Imager (SSM/I) on Defense Meteorological Satellite Program (DMSP) satellites, Advanced Microwave Scanning Radiometer-Earth Observing System (AMSR-E) on Aqua, and the Advanced Microwave Sounding Unit-B (AMSU-B) on the National Oceanic and Atmospheric Administration (NOAA) satellite series. The IR data, for the TMPA are collected by the international constellation of geosynchronous satellites. Additionally, data from TMI and the precipitation radar (PR) on TRMM is used as a source of calibration. The whole TMPA algorithm is constructed in four steps: (i) Microwave precipitation estimates are calibrated and merged. (ii) IR precipitation data are produced using the calibrated microwave results. (iii) Then, the microwave and IR precipitation estimates are combined filling missing data. (iv) Lastly, rain gauge data are incorporated for the final product. For a detailed explanation of the TMPA algorithm see Huffman et al. (2007).

A similar approach is used for the CMORPH product (CPC MORPHing technique, Joyce et al., 2004) which uses passive microwave information from SSM/I, AMSU-B, AMSR-E and TMI. The main difference to TMPA is that data gaps are treated differently by transporting rainfall features via spatial propagation information which is obtained from geostationary satellite IR data (Joyce et al., 2004).

PERSIANN (Precipitation Estimation from Remotely Sensed Information using Artificial Neural Networks, Sorooshian et al. (1998)) assimilates IR precipitation estimates from geosynchronous satellites. These estimates are then calibrated using microwave precipitation from low Earth orbit satellites. It differs from the other above described precipitation algorithms as its calibration technique involves an adaptive training algorithm that updates the retrieval parameters when microwave observations of precipitation become available (Sorooshian et al., 1998).

Inter-comparison studies show good agreement with ground based precipitation observations for these data products (e.g. Ebert et al., 2007; Novella and Thiaw, 2010; Romilly and Gebremichael, 2011; Liu et al., 2012; Pipunic et al., 2013; Pfeifroth et al., 2016, and references therein) which is, however, variable for different geographic regions, surface types and rain intensities.

In our study, we apply each precipitation product individually to differentiate between days with or without rain fall. From the comparison of the corresponding results we find that the uncertainties and differences among the precipitation data sets have only minor effects on the obtained results (see Appendix A, E).

## 2.4 Soil Moisture

The processes of nitrification and denitrification, which govern sNOx fluxes, are closely related to the soil water content (Meixner and Yang, 2006). In-situ measurements of soil moisture are sparse and difficult to extrapolate to broad geographic regions due to their highly heterogeneous nature. Combined satellite measurements of soil moisture overcome this issue by providing global coverage on a daily basis. Although the absolute value of soil moisture from merged satellite products has large uncertainties, relative variations triggered by precipitation events, should be evident in the time series.

Here, we use data from the Soil Moisture CCI (Climate Change Initiative) ECV project which merges level 2 soil moisture data derived from multiple satellite sensor products (Wagner et al., 2012) in order to construct a consistent long-time data set. Among the list of sensors that are included, are the C-band scatterometers on board of the ERS and METOP satellites and the multi-frequency radiometers SMMR, SSM/I, TMI, AMSR-E, and Windsat. The data sources include active (scatterometer) and passive (radiometer) microwave observations acquired preferentially in the low-frequency microwave range.

## 2.5 Other datasets

The datasets described above provide the basic information used in our research study. To evaluate and understand other influences on the retrieved trace gas levels, however, further atmospheric and environmental parameters are considered.

### 2.5.1 Lightning $NO_x$

Lightning represents a natural source of $NO_x$ in the upper troposphere, especially in the tropics (Bond et al., 2002) and, thus, has a potential impact on the measured $NO_2$ slant column densities. Estimates of lightning activity are captured by satellite instruments as well as ground-based stations like the World Wide Lightning Location Network (WWLLN, Holzworth et al., 2004; Rodger et al., 2006). WWLLN offers a continuous dataset which is based on 20–30 ground-based sensors that detect impulsive signals from lightning discharges, *sferics*, in the very low frequency (VLF) band (3–30 kHz) (Dowden et al., 2002). This algorithm is, thus, more sensitive to cloud-to-ground flashes because of their stronger radiation in the VLF band compared to intra-cloud flashes. In order to be classified as a lightning event the lightning strike must be detected by at least five stations. The detection efficiency (DE) varies to a large extent due to the spatial distribution of contributing stations. E.g. over Australia the DE is $\approx$ 80–90%, but only $\approx$ 10–20% over Africa (Rodger et al., 2006).

### 2.5.2 Fire activity

Biomass burning in specific regions is a major source of trace gases and aerosol particles (Crutzen and Andreae, 1990) and, therefore, must be considered in our analysis. The MODIS global monthly fire location product MCD14ML (Giglio et al., 2006) is used to filter out locations affected by fires.

### 2.5.3 Meteorology

In order to understand the prevailing meteorology and filter for special circumstances in the Sahel region, modelled data of soil and air temperature, pressure, humidity as well as wind fields are taken from the ECMWF ERA-Interim analysis (Dee et al., 2011). The model data is acquired at a spatial resolution of 0.25° and a temporal resolution of 6 hours over the period from 2007 to 2010. The data is publicly available via http://apps.ecmwf.int/datasets/.

### 2.5.4 Land Cover

The analysis of trace gas time series is also split up for different land cover types as they are related to different soil compositions and, thus, different $sNO_x$ potentials. Here, a land cover map for the year 2009 from the ESA initiated GlobCover project is used, which utilizes observations from the MERIS sensor on board the ENVISAT satellite mission with a spatial resolution of 300 m. The product is publicly available via http://due.esrin.esa.int/page_globcover.php and comprises 22 land cover classes defined with the United Nations (UN) Land Cover Classification System (LCCS) with an overall accuracy across all classes of 58% (Arino et al., 2007; Bontemps et al., 2011). The data is down-scaled using a most-common-value approach to identify dominant land cover types and to match the resolution of the other data sets. Thus, misclassifications might occur particularly over heterogeneous terrain and transition zones, while classification over homogeneous terrain is expected to be robust.

### 2.5.5 Water Vapour

Total column observations of $H_2O$ VCDs from GOME-2 give insight into the absolute humidity of the atmosphere at the time of the $NO_2$ observation from GOME-2 and, thus, a temporally more reliable estimate compared to modelled ECMWF data. $H_2O$ VCDs from GOME-2 are derived based on a DOAS retrieval using a $H_2O$ absorption band around 650 nm. Remaining non-linearities due to saturation effects are accounted for by a simple correction function determined from a radiative transfer model (RTM). Empirical AMFs are derived from the simultaneously measured $O_2$ absorption. Retrieval details and validation of the $H_2O$ VCDs can be found in Wagner et al. (2003, 2006) and Grossi et al. (2015).

## 3 Methodology

A daily global time series data set spanning from 2007 to 2010 for grid boxes of 0.25° x 0.25° is established comprising total precipitation and trace gas measurements. Level-2 products of the trace gases are screened for observations with effective cloud fraction above 20% and a solar zenith angle above 60°. Furthermore, observations coinciding with lightning or fire events on the same day and within the same grid box are filtered out.

The 3-hourly precipitation data is integrated over the 24 h period prior to the satellite overpasses of GOME-2, SCIAMACHY and OMI to collocate rainfall events and trace gas observations. For example, in the Sahel region, which is the main study region of this paper, the precipitation data are integrated from 13:30 UTC of the previous day to 13:30 UTC of the current day as this

corresponds to the local overpass time of OMI. This 24-hour period is called *Day* in the following pages (see Fig. 1). For the global analysis, the temporal integration is shifted by three hours in steps of 45° longitude.

As highest soil emissions are expected at the start of a wet season after a long drought phase of several weeks to months, the transition between dry and rainy seasons is the primary focus of this work. However, the length of drought phases are quite different for semi-arid areas in the world, varying from very long (several months in winter) in the Sahel to shorter periods (several weeks to months in summer) in South West Africa. Our approach considers grid boxes which experienced only little precipitation per day, *e.g.* < 2 mm, over a minimum number of days, *e.g.* 60 days, and a reasonable amount of precipitation on the first rain day (> 2 mm). Then, the trace gas column densities around this *first day of rainfall*, which is counted as *Day0* hereafter, are compared to the background levels during the preceding dry spell. The results vary slightly for different thresholds of the precipitation trigger, as shown in Appendix C. These sensitivity tests also show that a threshold of 2 mm/day leads to good statistics as well as representative responses in $NO_2$ VCDs.

Fig. 2 depicts a typical time series of precipitation (left panel) and $NO_2$ VCDs (right panel) for a five day period around the first rain event (on Day0) after a dry spell for a single grid pixel in the Sahel in April 2008. In the following, the days around the first day of rainfall are referred to as Day-3, Day-2, Day-1, Day0, Day+1, Day+2, Day+3 and so forth. It should be noted that there are almost no gaps in the precipitation time series, however, there are many in the trace gas time series. This is primarily due to the lower spatio-temporal coverage of trace gas products as well as the cloud, lightning and fire screening. In the example shown in Fig. 2 a 0.25° x 0.25° pixel is chosen which provides a complete $NO_2$ time series over 10 days. There is very little precipitation per day before the initial rain event. On Day0, precipitation exceeds a threshold of 2 mm, used to differentiate between 'rain' and 'no rain'. Investigating the time series of $NO_2$ around the first day of rainfall reveals a strong enhancement on Day0 and some smaller enhancement on Day-1 and Day1, whereas from Day-10 to Day-2 the $NO_2$ VCD is close to the pre-event level, i.e. the average $NO_2$ VCDs of Day-5 to Day-1. $NO_2$ VCDs after Day1 stay systematically higher than the background.

The time series for this single grid box represents the evolution of precipitation and trace gas VCDs around the first day of rainfall for a single grid pixel (experiencing first precipitation after an extended drought) demonstrating the basic principle of this study.

In order to achieve representative results with improved statistics, averaging the time series over many pixels is necessary. However, as we focus on pulsed soil emissions, averaging of time series from different pixels with rain events shifted in time has to be avoided. Furthermore, only a small subset of all possible pixels and their corresponding time series fulfils the conditions of the precipitation trigger. Thus, the individual time series are first synchronized in time relative to the first day of rainfall (Day0). The subsequent averaging method is applied, in the following section, either with focus on high spatial resolution or with focus on best statistics at the expense of losing spatial resolution by averaging over larger areas.

In the following sections a drought period of at least 60 days followed by a rain event (precipitation > 2 mm) is referred to as the reference case. The drought period of about two months is chosen as we find the highest response in $NO_2$ with this setting. In Appendix B, the impact of drought lengths on the derived soil emission pulses is investigated.

## 4  Results

### 4.1  Global Analysis

The algorithm described above, is applied to the full spatial extent covered by the TRMM/TMPA precipitation data set (-180° to 180° longitude, 50° to -50° latitude).

Fig. 3a displays the number of valid OMI observations on 1.25° x 1.25° grid pixels that fulfil the selection criteria, i.e. 60 days of drought and at least 2 mm of precipitation on Day0. For most regions in the world enough data points are found for our analysis; exceptions are regions with no pronounced seasonality in rainfall (e.g., tropical rainforests, North America, Europe) and regions where rain occasionally falls during the dry season (Southeast Asia). Our algorithm is not optimized for those regions.

The days prior to the first rain are assumed to represent a background level of $NO_2$. Fig. 3b depicts a background $NO_2$ map from OMI measurements obtained by averaging VCDs of the Day-10 to -2.

To examine variations in trace gas columns due to rain events, the enhancement of $NO_2$ VCDs on Day0 are considered with respect to to the background. Fig. 3c shows the spatial distribution of these absolute differences for OMI. In Fig. 3d, data points within two times the standard deviation, $\sigma$, of the background variation in the respective grid cell are screened out. Furthermore, as the uncertainty from spatial representativeness becomes the dominant uncertainty contribution if only few valid satellite pixels per grid cell are available (Boersma et al., 2016), at least 50% of possible data are requested per grid pixel in order to be considered in this analysis. The corresponding results for $NO_2$ VCDs observed by GOME-2 and SCIAMACHY are similar to Fig. 3d, but are more affected by noise due to poorer statistics (see Appendix D).

The most eminent features are the high enhancements of OMI $NO_2$ column densities on Day0 in the distinct band of the Sahel region around 15° N. Single grid pixels in this distinct band exceed absolute enhancements of $1 \times 10^{15}$ molec cm$^{-2}$ over background.

Similar enhancements in the Sahel were also observed by Hudman et al. (2012). In the south western part of Africa as well as over Australia spatially coherent enhancements are also present. Small scale, local enhancements are found also *e.g.* over India (also investigated by (Ghude et al., 2010)), regions nearby the Caspian Sea, the Middle East or China. An important finding is that there are no clustered reductions in $NO_2$ VCDs on Day0. Since we do not apply any land-sea mask, oceans serve as control regions for our algorithm: no significant differences in $NO_2$ are found over the vast majority of oceans area. However, over the Mediterranean sea and in proximity to coastal regions over oceans small-scale enhancements in $NO_2$ VCDs are detectable which might be related to advection.

The applied algorithm considers all data regardless of the season. Analysing the data based on different periods of the year reveals local enhancements in $NO_2$ VCDs in semi-arid areas matching dry-to-wet season transitions in these geographic regions (Fig. 4). In April/May/June, panel (a), the narrow band of the Sahel again is characterized by a mean enhancement of $NO_2$ of ~$1 \times 10^{15}$ molec cm$^{-2}$. This time period corresponds to the start of the rainy season in the Sahel after a long dry spell of 3–4 months. In September/October/November, panel (b), the strongest peaks in $NO_2$ VCDs are observed in South West Africa representing the start of the local wet season.

## 4.2 Detailed analyses over the Sahel region

The Sahel region represents a transition zone between the savannah in the south and the Saharan desert in the north. It is characterized by a strong seasonality in rainfall governed by the north-south movement of the Inter-Tropical Convergence Zone (ITCZ). The northward movement of the ITCZ starts in March and the northernmost position is reached, at 15° N, in August. In the following four months the Sahel region receives about 90% of its mean annual precipitation (Bell et al., 2000). The subsequent dry season begins in October and ends gradually with the start of the next wet season in April/May/June (AMJ-period).

Previous studies (e.g. Jaeglé et al., 2004; Hudman et al., 2012) argue that in this distinct geographic band pulsed soil emissions of $NO_x$, which can be detected from space, occur at the beginning of the wet season in spring. Our findings support these previous studies and delineate this narrow band from 10° to 18° N and from the West Coast of Africa essentially spanning the whole width of the continent, as shown in Fig. 4a. The pronounced $sNO_x$ features in the band of the Sahel during the AMJ-period enable a more detailed investigation of pulsed soil emissions.

We restrict our detailed analysis to the Central and Eastern part of the Sahel region (0-30° E, 12-18° N), similar as in previous studies (i.e. Jaeglé et al., 2004; Hudman et al., 2012). The western part of the Sahel shows a slightly weaker $NO_2$ response to rain pulses, which might be related to different inter-annual variability patterns and seasonal cycles of precipitation regimes (Lebel and Ali, 2009).

In order to depict the general behaviour of trace gas responses and to improve the statistics, four years (2007–2010) of the AMJ-period are averaged for the Eastern part of the Sahel. Fig. 5 depicts the evolution of multiple environmental variables around Day0 averaged over the study region. The precipitation amounts from the three different products, indicated in grey shades in each panel of Fig. 5 generally agree in their relative variation, showing little rain before Day0, a heavy rain event on Day0 and slightly higher precipitation after the first rainfall event compared to Day-10 to Day-1. The discrepancies among TMPA, CMORPH and PERSIANN products indicate that some rain events might be missed or assessed differently by the individual data products. Nevertheless, considering CMORPH or PERSIANN data as trigger leads to comparable responses in trace gases around the first day of rainfall (Appendix A).

The immediate wetting of the dry surface on Day0 is captured well by satellite observations of the volumetric soil moisture content as seen in Fig. 5a. After the initial wetting of the soil, the moisture content drops quickly during the following three days due to infiltration and evaporation. Similar behaviour is observed for total column densities of water vapour from GOME-2 in Fig. 5b. Water vapour content in the atmosphere gives insight into the ambient humidity and may indicate impending rain events, as humidity in the atmosphere typically rises prior to precipitation. On Days-10 to -2 water vapour steadily builds up in the atmosphere and peaks one day before the initial rain event. On Day0 and the following three days, the water vapour column densities drop, which is probably caused by, on the one hand, the removal of atmospheric water by precipitation and, on the other hand, by transport of dry air masses over the study area. The results for $NO_2$ VCDs using the three satellite instruments (OMI, GOME-2 and SCIAMACHY) show a consistent enhancement around the first day of rainfall (Fig. 5e). This points at a strong source linked to the dry-wet transition, i.e. the precipitation trigger. However, the magnitude of the enhancement varies

for the three instruments. The standard error of the mean (SME) value is indicated for each instrument and is generally smaller for OMI which has the best statistics.

The average absolute enhancement in the Sahel for GOME-2 $NO_2$ on Day0 compared to the background levels is about $5 \times 10^{14} \, \mathrm{molec \, cm^{-2}}$, whereas OMI and SCIAMACHY only observe an absolute enhancement of ~3 and $4 \times 10^{14} \, \mathrm{molec \, cm^{-2}}$, respectively.

The different magnitudes of the enhancements cannot be solely explained by differences in overpass time or pixel size as GOME-2 and SCIAMACHY are similar in both aspects. Furthermore, higher emissions are expected in the afternoon, i.e. at OMI overpass time, when the temperature is higher. The corresponding SCDs, however, (see Fig. 9e) indicate that the differences seen in the $NO_2$ VCDs are mainly caused by differences in the AMF calculation for the three data products.

Note that the enhancement is about $5 \times 10^{14} \, \mathrm{molec \, cm^{-2}}$ on average, while it was shown in the previous section that for single 1.25° boxes the absolute enhancements can be as high as ~$1 \times 10^{15} \, \mathrm{molec \, cm^{-2}}$. Smaller grid pixels show enhancements of up to ~$4 \times 10^{15} \, \mathrm{molec \, cm^{-2}}$, and for single events even larger enhancements are found).

Another striking feature, similar to the results for soil moisture, water vapour and precipitation, is the generally higher $NO_2$ VCDs during the ten days following the first rainfall event compared to the background levels before Day0. In sections 4.4 and 4.5, this important finding is studied more in detail by analyzing the $NO_2$ levels after Day0 depending on wind conditions and the precipitation on Day1 and beyond.

As indicated in the introduction chapter, HCHO emissions from soils were found in several laboratory and field experiments (e.g., Veres et al., 2014). In Fig. 5f we also analysed HCHO VCDs from OMI, SCIAMACHY and GOME-2 for potential pulsed emissions triggered by precipitation. The time series of HCHO for the three instruments, however, show no significant enhancement around the day of the first rain event. Possible reasons are the low-signal-to noise ratio for HCHO observations or very low emission rates.

## 4.3 Land cover analysis

Soils from different regions and land types are differently affected by precipitation and vary strongly in their microbial composition, nitrogen availability and pH values which presumably leads to strong differences in emission fluxes from soils. In the following study, the ESA GLOBCOVER land cover classification is used to characterize different land cover types. The data set is scaled down from the initial 300 m resolution to the 0.25° x 0.25° grid using a most-common-value method. The resulting land cover map is shown in Fig. 6a.

For different land cover types, both the $NO_2$ response on Day0 and the background level of $NO_2$ vary systematically, see panel (c). The observed $NO_2$ background VCDs per land cover type in the Sahel are mainly governed by biogenic emissions from soils and biomass burning. Anthropogenic activity and related emissions such as domestic fires or fertilized fields are at a very low level, and originate mostly from the southern, more populated part of the Sahel (Delon et al., 2010).

Systematic variations among the different land cover types are captured well: barren land, for example, shows lowest levels of $NO_2$ compared to all other land cover types. Barren land relates to deserts with very low nitrogen input resulting in low $sNO_x$, even after wetting. This land type is also associated with fewer rainfall events, see panel (b). Mosaic land covers (a mixture

between various vegetation types, grassland, cropland or forest), refer in this area to the loose term *savannah* delineating the transition zone between tropical forests and deserts. Savannahs can comprise various land cover sub-types and are characterized by distinct dry and wet periods with strong vegetation density and productivity during the wet season in summer. It is expected that savannah and cultivated land used for agriculture show strongest responses to initial rain events due to their higher potential for soil emissions. Panel (c) of Fig. 6 confirms these hypotheses: the largest $NO_2$ enhancements are found for cropland and savannah; grassland shows a significant, but smaller response; and the driest land cover type (bare area) shows only slightly enhanced $NO_2$ on Day0.

## 4.4 Influence from other sources on the $NO_2$ signal

In this section, we investigate the effects of possible additional sources of $NO_x$ such as fire or lightning and systematic errors in the satellite retrieval due to *e.g.* changes in cloud fraction. To minimize the influence of these effects, our algorithm excludes measurements where lightning, fires or an effective cloud fraction > 20% are detected.

### 4.4.1 Lightning $NO_x$

Lightning is a natural source of $NO_2$ in the upper troposphere (e.g. Schumann and Huntrieser, 2007, and references therein). Since lightning typically occurs in high convective clouds that may correlate with the first rain event, our analysis is potentially affected.

Fig. 7 depicts daily time series for $NO_2$ VCDs, precipitation and lightning counts averaged for the years 2007 to 2010. The seasonal evolution of the number of lightning strikes closely follows the precipitation patterns. Fig. 7 also illustrates that lightning is not a governing source of $NO_x$ in the Sahel as no correlation between lightning strikes and $NO_2$ VCDs can be found, although a direct proportionality would be expected. Precipitation also does not correlate well with the observed seasonal cycle in $NO_2$. This is, however, expected as microbial emissions of $NO_x$ from soils are not a linear function of soil moisture content or precipitation.

Fig. 8 shows results for the reference case, similar to Fig. 5, but with (solid lines) and without (dashed lines) lightning screening, i.e. grid pixels coinciding with a lightning event are removed. Because of the low detection efficiency (DE) of the WWLLN in African regions (~ 20%) the lightning screening is also tested for Central Australia (15–30° W, 2–10° S) where the DE of the WWLLN is very high (80–90%). Turning off the lightning screening leads to very similar results as for the reference case, but with a slightly stronger response in $NO_2$ VCDs on Day0 for all three instruments. While this enhancement might be partly caused by the additional $NO_x$ produced by lightning, also a larger number of true precipitation triggered events that lead to soil emissions may be included in this analysis. This is conceivable as clouds and thunderstorms accompanied by lightning strikes lead to the most heavy precipitation events. As the screening only causes minor changes in peak $NO_2$ columns, lightning can be excluded as the main cause of the observed $NO_2$ enhancements.

### 4.4.2 Fire

The seasonal cycle in fire counts, depicted in Fig. 7, shows highest activity in the Sahel in October and November for the years 2007 to 2010, while average $NO_2$ VCDs are highest in summer.

Switching off the routine data screening for pixels that coincide with fire events in the same 0.25° x 0.25 ° grid pixel results in no change of the $NO_2$ signal (not shown). This is due to the fact that only very few fires occur in the wet season, on average only in 0.002% of all individual time series on Day0 in the reference case, excluding fire as an important $NO_2$ source within our analysis.

### 4.4.3 Cloud effects

We have investigated possible cloud effects on our results by analyzing the temporal evolution of the mean cloud fraction (CF), $NO_2$ VCDs, and $NO_2$ SCDs around the precipitation event. The latter was added as it provides the actual measured signal without involving a tropospheric AMF, which is generally very sensitive to clouds.

Fig. 9a depicts the mean effective cloud fractions (CF) for SCIAMACHY, GOME-2 and OMI for the Sahel region. The differences of the absolute value of the CF are probably related to the different cloud algorithms between GOME-2/SCIAMACHY (FRESCO+) and OMI (OMCLDO2). The different temporal variation might also be partly related to the different overpass times and pixel sizes among the three satellite instruments. From these results we conclude that the observed $NO_2$ peaks around Day0 are not caused by cloud effects for the following reasons: First, for all sensors only small cloud fractions (<11%) are found (for measurements with CF < 0.2). Second, for SCIAMACHY and GOME-2 observations no systematic temporal variation of the CF is found. Third, the small but systematic enhancement of the CF around Day0 found in the OMI observations would rather lead to a decrease (due to the shielding effect) of the $NO_2$ SCDs around Day0 as soil emissions are expected to remain close to the surface. If only measurements with cloud fractions >20% are considered, no spike is observed in the SCDs (Fig. 9f); GOME-2 and OMI even show a dip on Day0, which is also seen in the respective VCDs (Fig. 9d). Interestingly, while there is a strong systematic enhancement of the FRESCO+ and OMCLDO2 cloud fractions, the $NO_2$ SCDs show no peak around Day0 for GOME-2 and OMI. This indicates that clouds effectively shield the pulsed soil emissions.

### 4.4.4 Influence of transport processes

Finally the possible influence of transport processes, which might be correlated with the occurrence of the first rain event, is investigated. As depicted in Fig. 10, a strong southerly wind is blowing at ground level (1000 hPa) the two days before the first rain event and on Day0 in the Sahel. In order to investigate whether polluted air from southern locations, especially the Tropics, is transported northward into the Sahel, we repeat the analysis for days governed by either northerly or southerly winds (Fig. 11). For the distinction between both directions we require that wind vectors from ECMWF at three different altitudes (600, 850 and 1000 hPa) point to the same direction in either case. The left panel in Fig. 11 depicts results for northerly winds; the right panel for southerly winds. Although the background levels of $NO_2$ are reduced on days with northerly winds, enhancements in VCDs around the first day of rainfall remain apparent despite low statistics, especially for OMI and GOME-2

observations. For days with southerly winds the background is slightly higher and clear spikes for the OMI and GOME-2 observations can also be detected. Hence, these findings indicate that atmospheric transport has a systematic influence on the background $NO_2$ levels (see also Section 4.5), but not substantially on the enhancement around Day0.

### 4.5 Latitudinal background correction and emissions after Day0

In the reference case analysis presented in the sections above, $NO_2$ VCDs show an enhancement with respect to the background, i.e. the average $NO_2$ VCDs before Day-1, not only on Day0, but also on Day-1, Day2 and Day3. It is assumed that the *background* $NO_2$ VCDs are not influenced by the precipitation-triggered $sNO_x$ pulsing event and, thus, can be used as reference to derive an absolute enhancement in $NO_2$ VCDs predominantly induced by pulsed soil emissions. However, it is shown in Fig. 5e and Fig. 9c,e that the $NO_2$ VCDs after the pulsing event (Day3 to Day10) are consistently higher compared to the

background before Day0. This could be related to inflow of soil $NO_x$ from adjacent pixels which receive the first rain shortly after. However, initial tests showed that the number of such incidents is quite small (less than 5%). For this reason, and because the effect is possibly offset by sNOx advection out of the pixel of interest, it is assumed that the effect of inflow is not dominant. As we focus on events with strong emissions (compared to background), a "smearing effect" by advection leads to a consistent underestimation of the peak and subsequent emissions. Thus, we conclude that inflow cannot explain the enhancement for

14 days after the first rain event. Furthermore, it remains unclear to what extent the enhanced $NO_2$ VCDs are affected by a possible underlying change in the background.

   This leads to four interesting questions: (i) are the slightly enhanced $NO_2$ VCDs after Day0 related to the pulse on Day0? (ii) In case the enhanced $NO_2$ VCDs after Day0 are (partly) caused by other sources: what effect does a background correction have on the retrieved absolute enhancements in $NO_2$ VCDs around Day0? (iii) Is continuous precipitation the cause for the

enhancement after Day0? (iv) In case the enhanced $NO_2$ VCDs after Day0 are only related to the pulsed rain event: can we give quantitative estimates on these 'continuous' soil emissions?

   Fig. 12a depicts the reference case analysis for the Sahel region with respect to 120 days around the first day of rainfall after the drought period. The $NO_2$ VCDs observed by the three satellite instruments show consistent patterns in the spike around Day0 and still slightly enhanced $NO_2$ VCDs 60 days after Day0.

To investigate whether the increased $NO_2$ VCDs after Day0 are related to the pulsed rain on Day0 or caused by a general change of the background $NO_2$ VCDs (*e.g.* related to a seasonal variation), we try to estimate the temporal evolution of the background (not affected by a pulsed rain event) in the following.

   In a first attempt $NO_2$ VCDs are averaged over all grid pixels located at the same latitude as the identified rain events assuming latitudinal homogeneity in $NO_2$ background VCDs. This assumption is justified in the Sahel due to the latitudinal

distribution of its land cover types governing $NO_2$ VCDs. The corresponding results are depicted in Fig 12b). While compared to Fig 12a) a much smoother temporal evolution is found (because more data is averaged), but apart from the much smaller spike on Day0, very similar values can be seen in both panels. This confirms our assumption that the main part of the increase of the background value is not caused by the precipitation on Day0. However, the spike in $NO_2$ VCDs around Day0 is still evident because this averaging method still considers the initial pixel with its adjacent neighbours which are probably affected

by either the overall precipitation pattern or spatial aliasing effects during the gridding of the $NO_2$ data products. Thus, in Fig 12c) a 10 pixel buffer is additionally applied to the algorithm. Screening out such pixels leads to a time series of $NO_2$ without the distinct spike around Day0.

The time series of $NO_2$ VCDs retrieved using the two latitudinal averaging methods is denoted as the prevailing background and is subsequently subtracted from the reference case analysis (Fig. 12). In the first case, without the application of an additional buffer (Fig. 12d), absolute enhancements of $0.43 \times 10^{15}$ molec cm$^{-2}$ for GOME-2 and SCIAMACHY are found on Day0. Also a steady increase in $NO_2$ VCDs several days prior to Day0 is observed. Although the pronounced spike in $NO_2$ VCDs decreases rapidly in the days following Day0, it lasts several weeks until the VCDs reach a minimum (but stays still slightly higher than on Day-60 to Day-20). A similar behaviour is observed for the case study with buffer screening (Fig. 12e). Here, absolute enhancements of $0.4 \times 10^{15}$ molec cm$^{-2}$ for OMI and $0.62 \times 10^{15}$ molec cm$^{-2}$ for GOME-2 and SCIAMACHY are found on Day0.

From these results we conclude that the slightly enhanced $NO_2$ VCDs after Day0 are related to the precipitation on Day0 at or close to the considered location. As our focus is on the quantification of the emission pulse triggered by the first rain of the wet season, it still needs to be clarified whether the enhanced $NO_2$ VCDs after Day0 are induced by the initial precipitation on Day0 or by continuous precipitation during the following days. To answer this question we extract time series with the additional selection criterion of 3, 5, 10 and 20 days of no precipitation following Day0. Fig. 13a depicts the OMI $NO_2$ VCDs for 0, 3, 5, 10 and 20 days without precipitation after Day0. The intercomparison of these time series is not straightforward as the background values vary for each case indicating that the time series are captured at different dates throughout the April-May-June period. Longer drought periods after Day0 are more likely at the very beginning of the wet season (April), whereas more constant rainfall dominates at a later stage, e.g. in June. In this time period background $NO_2$ VCDs generally increase in the Sahel. For this analysis we assume that the impact of the different dates only affects the background and not the enhancement on Day0 as the selection criteria (precipitation and drought length thresholds) presumably have the largest effect. In order to analyse the enhancement around Day0 only, we apply the above described latitudinal background correction to each time series individually which reduces the influence of the background drastically. Fig. 13b,c show the corresponding background $NO_2$ VCDs derived without and with the aforementioned buffer screening applied. Naturally, the absolute change between the different background time series is smaller compared to panel (a) as they represent averages over all pixels on the same latitude. The changes on panel (c) are even smaller because the pixels in proximity to the triggered rainy pixel are excluded from the averaging. Finally, Fig. 13d,e depict the differences between panel (a) and panels (b) and (c), respectively. As expected, the enhancements in $NO_2$ VCDs around Day0, are more pronounced for cases including a 10 pixel buffer during the background correction. Although the absolute enhancement on Day0 is almost identical for the different cases, the time series still differ in the observed background $NO_2$ VCDs. These systematic differences, seen in panels (e) and (d), hint at more complex variations in the background which are not entirely resolved by our correction. They could also be related to the fact that for the cases with longer dry periods after Day0 the probability of precipitation in the vicinity of the considered location is lower. This implies, again, also differences in space and time of the observations which influences the observed $NO_2$ VCDs.

Generally, we find that the NO$_2$ VCDs stay enhanced after Day0 for a period of about two weeks, almost independent from the duration of the dry period after Day0. This gives rise to the assumption that the enhanced emissions are mostly caused by the initial rain event on Day0. Here, only results for OMI are presented because it has best statistics. Nevertheless, the analysis with data from the other instruments leads to the same conclusion.

## 4.6  Further study regions

As could be seen in the global analysis in Fig. 3, large scale hot-spots in NO$_2$ VCD enhancements are not only detectable in the Sahel, but also in South West Africa and in Australia. Subsequently, we present also the average NO$_2$ VCDs around the first day of rainfall for all three satellite instruments in these two regions.

Fig. 14a depicts the results for South West Africa (17–23° E, 22–28° S) for a drought period (precipitation < 2 mm) of at least 60 days in the months September/October/November 2007–2010 representing the transition phase between the dry summer and following wet season. Compared to the Sahel reference case, the evolution of NO$_2$ VCDs before and after the first day of rainfall increases and decreases more gradually without having a distinct spike on Day0. This might be due to different environmental conditions such as soil type or lower statistics because of the much smaller spatial extent. It is, thus, more difficult to estimate the absolute enhancement compared to a defined background level. The difference between highest and lowest NO$_2$ VCDs in the full time series is ~$0.5 \times 10^{15}$ molec cm$^{-2}$ for all three instruments.

Fig. 14b shows the analysis results for NO$_2$ VCDs for the central part of Australia (120–145° E, 22–31° S) for the time series from 2007 to 2010. Since the seasonality in rainfall in this region is less pronounced, the full time series is considered in the analysis. The well pronounced spikes shows an absolute enhancement of ~$0.3 \times 10^{15}$ molec cm$^{-2}$ for the three instruments, which is comparable to the findings from the Sahel and South West Africa.

## 5  Discussion

### 5.1  Estimated nitrogen emission fluxes from the emission pulse

Soil emissions of trace gases are not only limited to the specific days we investigated in this study (based on the selection criteria for the temporal evolution of precipitation), but can play an important role in the atmosphere during specific seasons and throughout the whole year (Steinkamp et al., 2009). Global chemistry models considered the mean seasonal behaviour of soil emissions in the past, but they were insensitive to rapid changes on a daily basis, *e.g.* during the onset of the wet season. Pulsed emissions of sNO$_x$ have been recognized to be a significant short term local enhancement which can be parametrized in GCMs as shown by Hudman et al. (2012) for the GEOS-CHEM model. The latter study investigates pulsed soil emissions of NO$_x$ in the Sahel region and finds a comparable magnitude (49% relative increase of OMI NO$_2$ VCDs on Day0) and length of the pulsing event (1-2 days) following the first rainfall. Here, we provide further evidence supporting that and other reported studies, i.e. the observed enhancements in NO$_2$ VCDs are consistent for multiple instruments and are not introduced by retrieval errors or interfering sources.

For the Sahel region we find significant mean enhancements on the first day of rainfall in April-May-June averaged over four seasons (2007–2010) of ~ $4 \times 10^{14}$ molec cm$^{-2}$, as observed by OMI, after the prolonged dry spell. However, we see much stronger enhancements for single pixels on the original fine resolution grid (0.25°) of up to ~$4 \times 10^{15}$ molec cm$^{-2}$. Considering these values as upper and lower limits for the sNO$_x$ enhancement on Day0, we can estimate emission fluxes. The top-down emission flux for NO$_x$ can be inferred from the NO$_2$ VCD by mass balance: E = $\Omega_{NO_2}/\tau_{NO_2}$ with $\Omega_{NO_2}$, being the tropospheric NO$_2$ VCD and $\tau_{NO_2}$ the lifetime. The lifetime for $\tau_{NO_2}$ is mostly determined by the oxidation of NO$_x$ to HNO$_3$ in the boundary layer and typically ranges between 4 and 10 hours in the tropics (Martin, 2003). Consequently, we find that the emission flux of nitrogen (N) for pulsing events, considering an assumed NO$_2$ life time of 4 hours, in the Sahel is between 6 ng N m$^{-2}$ s$^{-1}$ for a conservative estimate and 65 ng N m$^{-2}$ s$^{-1}$ on the upper limit on Day0. This is in line with findings from Jaeglé et al. (2004) who give an estimate of 20 ng N m$^{-2}$ s$^{-1}$ for rain-induced sNO$_x$ pulses in June in the Sahel. Furthermore, field studies suggest emission fluxes of nitrogen of about 2–60 ng N m$^{-2}$ s$^{-1}$ (Johansson and Sanhueza, 1988; Davidson, 1992a; Levine et al., 1996; Scholes et al., 1997) which covers our estimated upper and lower limits. Previous studies also note the strong dependence of soil emissions on temperature. We conducted initial tests using ECMWF soil temperature data (see Appendix F), but found no clear temperature dependence of sNO$_x$ emissions. This is probably due to the systematic relation between temperature and the seasonal cycle which affects the spatio-temporal selection of the data, e.g. in April more southern pixels are selected and in June more northern pixels. In consequence, we indicate the need for more detailed investigation on how these pulsed emissions are affected by different soil temperatures.

After the first direct emission pulse, emissions remain at an enhanced level for a period of about two weeks as described in the previous chapter with approximately 3.3 ng N m$^{-2}$ s$^{-1}$ averaged over the Sahel. Analogous to the pulsed emission on Day0, this value probably represents a lower limit because of the rather low spatial resolution of the extracted time series. These emissions are almost independent from the period of dry days after Day0 which indicates that they are potentially caused by the initial rain on Day0 and not by subsequent precipitation the following days. Possible advection effects into or out of the considered pixels may thereby raise or lower the retrieved sNO$_x$ fluxes. The analysis based on different dry phases after Day0 also changes the probability for first rain events in pixels in close proximity. As we do not find strong differences in the emissions for different dry phases after Day0, we conclude that the inflow of NO$_x$ from adjacent pixels is not the dominant source for the enhancement after Day0. In contrast, a systematic underestimation of the emissions is likely due to advection out of the central pixel. In sum, we estimate that the integrated emissions after Day0 are potentially of the same order of magnitude as those from the first emission pulse on Day0.

Peak emissions of sNO$_x$ pulses typically occur on the scale of 1-3 days (Kim et al., 2012) in accordance to our results for the Sahel, South Africa and Australia showing peak emissions shortly after the first re-wetting. Some studies, on the other hand, measure peak emissions several days after the first re-wetting of the soil, *e.g.* seven days as observed in field by Oikawa et al. (2015). Our algorithm does not specifically distinguish between such cases by taking average time series after the first precipitation event. Single pixels within the regions we investigated may exhibit peak emissions several days after the initial precipitation which would, however, not be resolved by our analysis.

Our study focuses on the quantification of pulsed soil emissions and determines the $NO_2$ enhancement on Day0 and the following days with respect to a sophistically determined background. However, the seasonal pattern of the determined background, i.e. the $NO_2$ enhancement at the onset of raining season (compare Jaeglé et al., 2004), clearly indicates that it is mainly driven by microbial emissions from soils as well: from pulsed emissions discarded by our strict selection criteria or continuous emissions during wet season. Note that the seasonal pattern of $NO_2$ over the Sahel, as shown in Fig. 7, can neither be explained by biomass burning nor lightning. Fig. 13c of the manuscript shows that the background $NO_2$ VCDs are about $0.9 * 10^{15}$ molec cm$^{-2}$. This is about $0.17 * 10^{15}$ molec cm$^{-2}$ higher than background in winter. Thus, in addition to the pulsed emissions quantified above, a mean background of $0.17 * 10^{15}$ molec cm$^{-2}$ can be attributed to soil emissions as well. These estimates are based on TMPA precipitation data. For other precipitation products (CMORPH or PERSIANN), results change only slightly (see Appendix E).

In summary, we discriminate between soil emissions within: (a) 1-3 days (initial peak), (b) 14 days, and (c) several months (background during the wet season). The separate quantification of soil emissions belonging to these three categories might also be adopted in model parametrizations of soil emissions. However, further research needs to be conducted on how these emission categories vary for different regions worldwide.

## 5.2 Seasonal soil nitrogen emissions in the Sahel

In this section we quantify the total soil emissions, both due to pulsed emissions and background, for the Sahel region. For the pulsed emissions on Day0 (category a) and the following 2 weeks (b), the fluxes estimated above are multiplied by the area of the investigated region (0-30°E, 12-18°N). The statistics of our analysis in the Sahel suggest that on average one large pulsing event (after 60 days of drought) occurs within a single pixel in the April-May-June period. Scaling up the Day0 emissions results in 1.2 GgN and 12 GgN, considering the lower and upper flux estimates estimated above. Analogously, the emissions over the following two week period add up to 8.8 GgN. Together this sums up to 10.1-20.8 GgN emissions due to pulsing. As mentioned above, the observed increase of the background in the AMJ-period of $0.17 * 10^{15}$ molec cm$^{-2}$ is mainly driven by microbial emissions from soils as well. When integrated over the complete April-May-June period, these seasonal soil emissions correspond to 46.4 GgN (again based on a $NO_x$ lifetime of 4 hours). Consequently, the emissions due to pulsing contribute about 21-44% additionally to seasonal soil emissions for the Sahel and dominate the local $NO_x$ concentrations on the particular days.

Jaeglé et al. (2004) determine top-down total soil emissions from GOME-2 measurements of about 400 GgN for North Equatorial Africa (0-18°N) in June alone. Our estimated total soil emissions of nitrogen (56.5-67.2 GgN for AMJ) are smaller, but are determined for a smaller region as well which makes a direct comparison difficult.

## 5.3 Enhancements in $NO_2$ VCDs on Day-1

For all analysed study regions and individual grid pixels showing significant $NO_2$ spikes on Day0, we also find enhancements in the $NO_2$ VCDs one to two days before the first day of rainfall. The phenomenon is especially pronounced for the study region in South West Africa. The finding of enhanced $NO_2$ VCDs before Day0 stands in contrast to the general expectation

that soil emissions, *e.g.* of $NO_x$, are only caused by the initial rain event and the subsequent wetting of the soil. However, absolute humidity shows a steady increase several days before the first rain event. Thus, reasons for the early increase of $NO_2$ VCD may be an increase in atmospheric moisture content and dew-fall, a misclassification of rainfall intensity by the precipitation algorithms, transport of $NO_2$ from neighbouring regions or spatial aliasing effects during the gridding of the satellite observations, i.e. the overlap of the ground footprint onto multiple grid boxes of the precipitation products. The latter two can occur if the ground pixel observed by the satellite overlaps with two or more grid pixels of the precipitation products or vice versa. This error is difficult to estimate, but should be more pronounced for larger satellite ground pixels, i.e. from SCIAMACHY, and less for instruments with smaller footprints, i.e. OMI. The fact that all instruments observe similar enhancements already on Day-1 indicates that this possible error source is not the dominant cause for the early increase in $NO_2$ VCDs. The use of three instruments for detecting $NO_2$, each having different overpass times during the day, also makes it less likely that a temporal mismatch of the precipitation and trace gas products, as described in the Methodology section, leads to the enhancements on Day-1.

In Fig. 5b it is shown that water vapour ($H_2O$) VCDs in the atmosphere retrieved by GOME-2 increases continuously for 10 days before the first precipitation event and peaks on Day-1. We speculate that the moist air over the extremely dry top soil layer induces initial $sNO_x$ emissions, despite the fact that the soil is not directly wetted by rain. Also an enhanced dew formation and water adsorption potential, which both are important sources of water in semi-arid areas have a major effect on microbiological activity (Verheye, 2008). The probability for nightly condensation over drying soils increases with higher absolute humidity. Although observations of these quantities are sparse, measurements in Israel, Jordan and in South Africa hint at contributions of 12 to 40 mm water per year in semi-arid regions (Verheye, 2008; Nicholson, 2011, and references therein). Transport of polluted air from the Tropics northwards presumably also leads to enhanced $NO_2$ VCDs before Day0. However, this effect is expected to be quite low as the enhancements before Day0 are also seen for cases dominated by northerly winds coming from the Sahara which are generally associated with lower $NO_2$ VCDs (see Fig. 11).

## 6 Conclusions

We have presented a top-down approach to infer rain-induced emission pulses of $NO_x$ based on space-based measurements of $NO_2$. This is achieved by synchronizing time series at single grid pixels according to the first day of rain after a dry spell of prescribed duration. The method is applied globally and provides constraints on pulsed soil emissions of $NO_x$ in regions where the $NO_x$ budget is seasonally dominated by soil emissions. This approach is similar to Hudman et al. (2012), but extended by (a) performing the analysis globally with (b) high spatial resolution, and (c) keeping full track of the temporal evolution several weeks before and after a rain pulse with daily resolution. The latter was used to (d) perform a sophisticated background correction, which turned out to be necessary in order to account for the seasonal variations in the time series and allows to (e) quantify rain-induced soil emissions also beyond the strong peak on the first day of rain.

Sensitivity studies were conducted in order to (i) evaluate the impact of the a-priori assumptions on thresholds for daily rainfall, i.e. the amount of precipitation and the required duration, (ii) investigate to what extent other $NO_x$ sources like

biomass burning or lightning $NO_x$ might interfere, and (iii) carefully check for possible retrieval artefacts (*e.g.* caused by clouds). None of these effects has shown to be critical for our conclusions.

Note, however, that our method was optimized for the quantification of pulsed soil emissions from space by demanding long droughts and good viewing conditions (low cloud fractions) on the day of precipitation onset. Thus, regions showing no clear response for these strict selections might still be capable of rain-induced soil emissions.

Strong peaks of enhanced $NO_2$ VCDs on the first day of rainfall after prolonged droughts are found in many semi-arid regions, in particular in the Sahel, South-West Africa, Australia and parts of India. A similar analysis for HCHO VCDs showed no indication for pulsed soil emissions. Closer inspection of the Sahel shows a strong dependence of precipitation-induced $NO_x$ emissions on land type cover. This finding confirms similar results from laboratory measurements.

For the Sahel region, absolute enhancements of the $NO_2$ VCDs on the first day of rain based on OMI measurements 2007-2010 are on average $4 \times 10^{14} \, \mathrm{molec \, cm^{-2}}$ and exceed $1 \times 10^{15} \, \mathrm{molec \, cm^{-2}}$ for individual grid cells. Results for SCIAMACHY and GOME-2 are comparable, and the slight differences can be primarily explained by different footprints, overpass times, cloud products, and retrieval schemes. Assuming a $NO_x$ lifetime of 4 hours, this corresponds to soil $NO_x$ emissions in the range of $6 \, \mathrm{ng \, N \, m^{-2} \, s^{-1}}$ up to $65 \, \mathrm{ng \, N \, m^{-2} \, s^{-1}}$ on Day0, in good agreement with literature values. Apart from the clear first-day peak, $NO_2$ VCDs are moderately enhanced ($2 \times 10^{14} \, \mathrm{molec \, cm^{-2}}$) compared to background over the following two weeks suggesting potential further emissions during that period of about $3.3 \, \mathrm{ng \, N \, m^{-2} \, s^{-1}}$. With respect to the seasonal $NO_x$ budget, we assess a contribution between 21 to 44% from these rain-induced intense pulsing events to total soil $NO_x$ emissions in the Sahel.

In conclusion, our findings facilitate a detailed characterization and estimation of emission budgets for intense $sNO_x$ pulses, triggered by individual rain events, which can improve parametrizations in modelling studies by dividing soil emissions into several parts: (i) pulsed emissions on short time scales, (ii) enhanced emissions after the initial pulse, and (iii) seasonal background emissions.

*Acknowledgements.* The authors wish to thank the World Wide Lightning Location Network (http://wwlln.net), a collaboration among over 50 universities and institutions, for providing the lightning location data used in this paper. We acknowledge the free use of tropospheric $NO_2$ and HCHO column data from the GOME-2, SCIAMACHY and OMI sensors from http://www.temis.nl and would like to thank Isabelle De Smedt for providing the most recent version of the HCHO products. We thank also all other authors of data products used in this study for their efforts in producing and providing their data. Furthermore, we would like to thank Bettina Weber, Buhalqem Mamtimin and Franz X. Meixner for the inspiring and constructive discussions.

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

**Table 1.** List of acronyms for commonly used instruments and products in this paper

| Short Name | Long Name | References |
|---|---|---|
| CMORPH | CPC (Climate Prediction Center) MORPHing technique | Joyce et al. (2004) |
| DOMINO | Derivation of OMI tropospheric $NO_2$ project (v2.0) | Boersma et al. (2011) |
| ECMWF | European Centre for Medium-Range Weather Forecasts | Dee et al. (2011) |
| FRESCO+ | Fast Retrieval Scheme for Clouds from the Oxygen A band | Wang et al. (2008) |
| GOME | Global Ozone Monitoring Experiment | Burrows et al. (1999) |
| GOME-2 | Global Ozone Monitoring Experiment–2 | Callies et al. (2000); Munro et al. (2006, 2015) |
| MERIS | MEdium Resolution Imaging Spectrometer | Rast et al. (1999) |
| MODIS | Moderate Resolution Imaging Spectroradiometer | Kaufman et al. (1998) |
| OMCLDO2 | Cloud Pressure and Fraction using $O_2$-$O_2$ absorption | Acarreta et al. (2004) |
| OMI | Ozone Monitoring Instrument | Levelt et al. (2000, 2006) |
| PERSIANN | Precipitation Estimation from Remotely Sensed Information using Artificial Neural Networks | Sorooshian et al. (1998) |
| SCIAMACHY | SCanning Imaging Absorption spectroMeter for Atmospheric CHartographY | Bovensmann et al. (1999) |
| SMMR | Scanning Multi-channel Microwave Radiometer | Gloersen and Barath (1977) |
| SSM/I | Special Sensor Microwave Imager | Wentz and Spencer (1998); Wentz (2013) |
| TMI | TRMM Microwave Imager | Wentz et al. (2001) |
| TMPA | Tropical Rainfall Measuring Mission Multisatellite Precipitation Analysis | Huffman et al. (2007) |
| TRMM | Tropical Rainfall Measuring Mission | Simpson et al. (1996); Huffman et al. (2007) |
| TROPOMI | TROPOspheric Monitoring Instrument | Veefkind et al. (2012) |
| WWLLN | World Wide Lightning Location Network | Dowden et al. (2002); Holzworth et al. (2004); Rodger et al. (2006) |

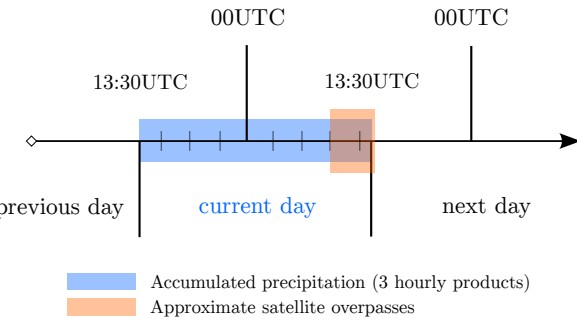

**Figure 1.** Schematic of the 24-hour time period selected for the integration of precipitation data. The eight 3-hourly precipitation rates prior to the overpass times of the SCIAMACHY, GOME-2 and OMI satellite sensors are summed.

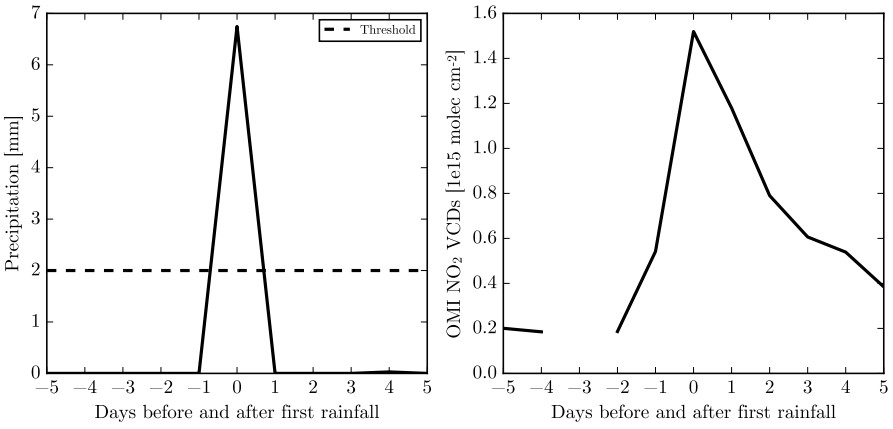

**Figure 2.** Time series of TMPA precipitation (a) and OMI $NO_2$ VCDs (b) for a ten day period around the first rain event for a single grid pixel in the Sahel on 11/04/2008 at 15.25° N, 25.5° E. A threshold of 2 mm precipitation per day is chosen and at least 60 days of drought are required.

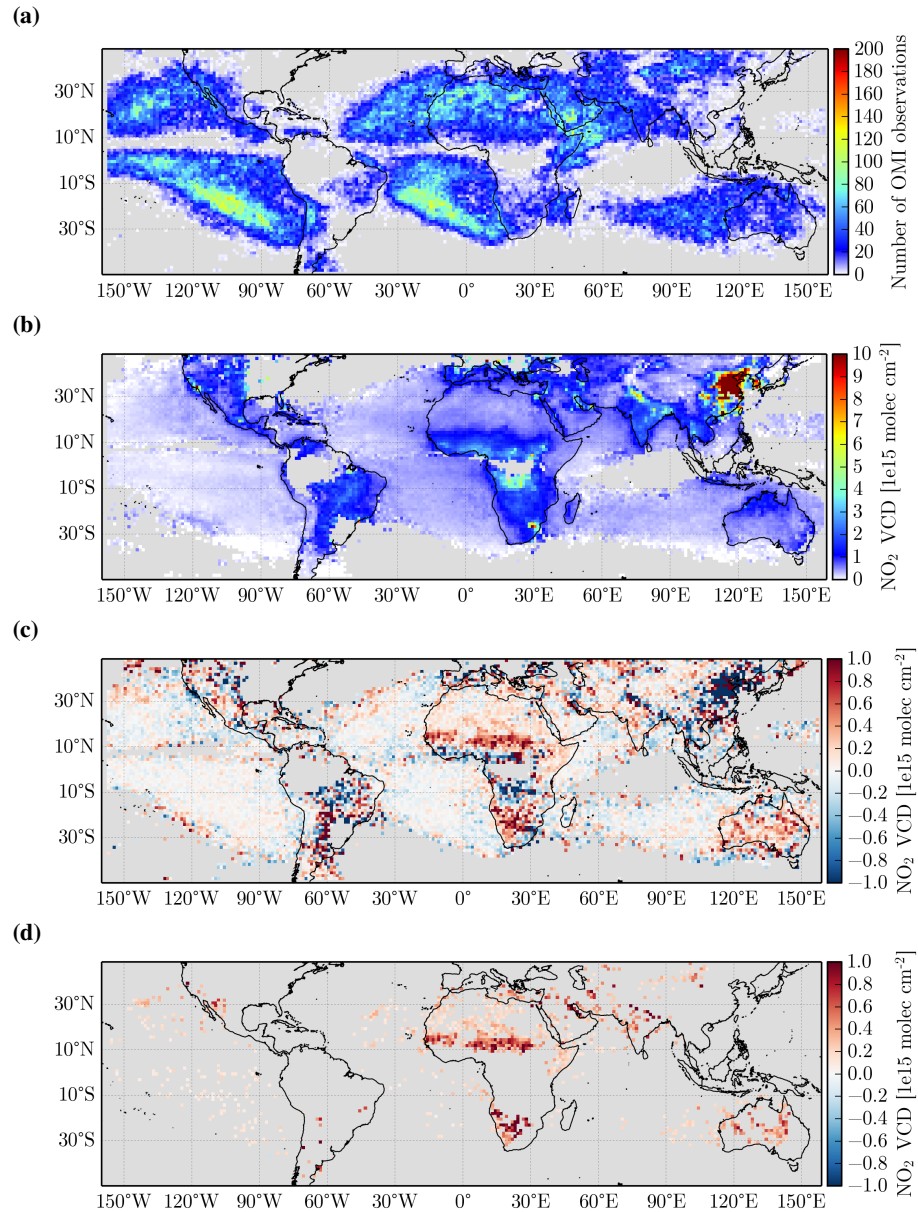

**Figure 3. (a)** Number of valid measurements per grid pixel on Day0. **(b)** OMI $NO_2$ background levels averaged for days -10 to -2 before the first rain event after 60 days of drought for each pixel ($0.25°$ lat/lon) and then averaged for boxes of $1.25°$ lat/lon. **(c)** OMI $NO_2$ VCD absolute differences on Day0 (first day of rainfall) compared to Days -10 to -2. Reductions in $NO_2$ VCDs on Day0 depicted in blue colours, enhancements in red. **(d)** as (c) but screened for significant changes (see text). Extensive enhancements over the Sahel and South Africa are evident. Pixels containing less than 20 measurements on Day0 (or less then 50 measurements from Day-10 to Day-2) are screened out.

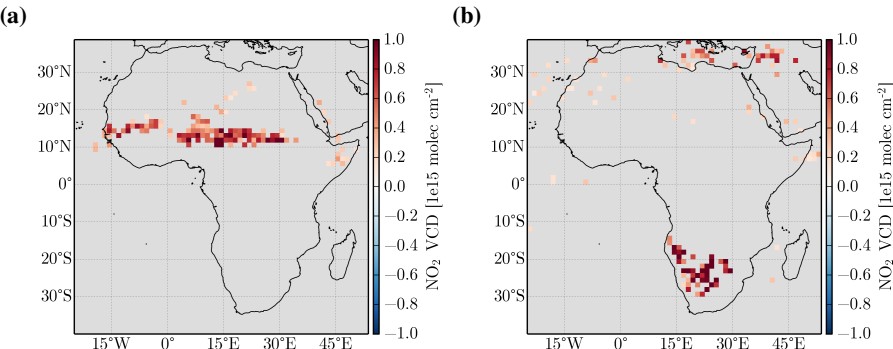

**Figure 4. (a)** Significant OMI $NO_2$ VCD enhancements (in $1\times10^{15}$ molec cm$^{-2}$) on Day0 compared to the background level for April-May-June (2007–2010) which represents the start of the wet season after the dry period in the Northern part of Africa. Extensive enhancements in $NO_2$ over the narrow band of the Sahel can be seen. **(b)** The same for September/October/November (2007–2010), whereby strong enhancements are found in South West Africa. This time period reflects the transition time between the dry and wet season in this region.

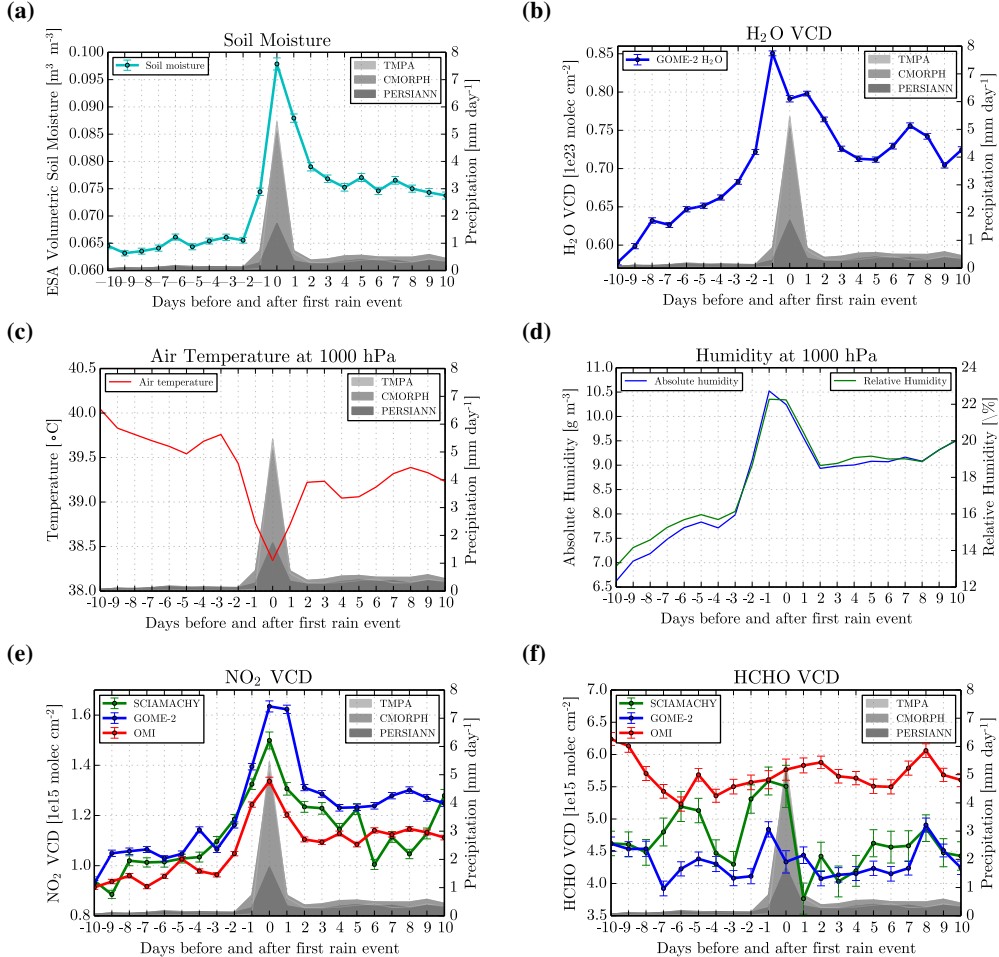

**Figure 5.** Temporal evolution of several quantities around the day with the first rain event for the Sahel region after at least 60 days of drought for April/May/June (2007–2010). Grey shaded areas represent precipitation estimates from TMPA, CMORPH and PERSIANN. **(a)** Blended ESA-CCV soil moisture. **(b)** Water vapour total column densities from GOME-2. **(c)** Temperature at 1000 hPa from ECMWF Interim Analysis. **(d)** Relative and absolute humidity at 1000 hPa from ECMWF Interim Analysis. **(e)** $NO_2$ VCDs from SCIAMACHY, GOME-2 and OMI with standard mean error (SME). **(f)** HCHO VCDs from SCIAMACHY, GOME-2 and OMI with SME.

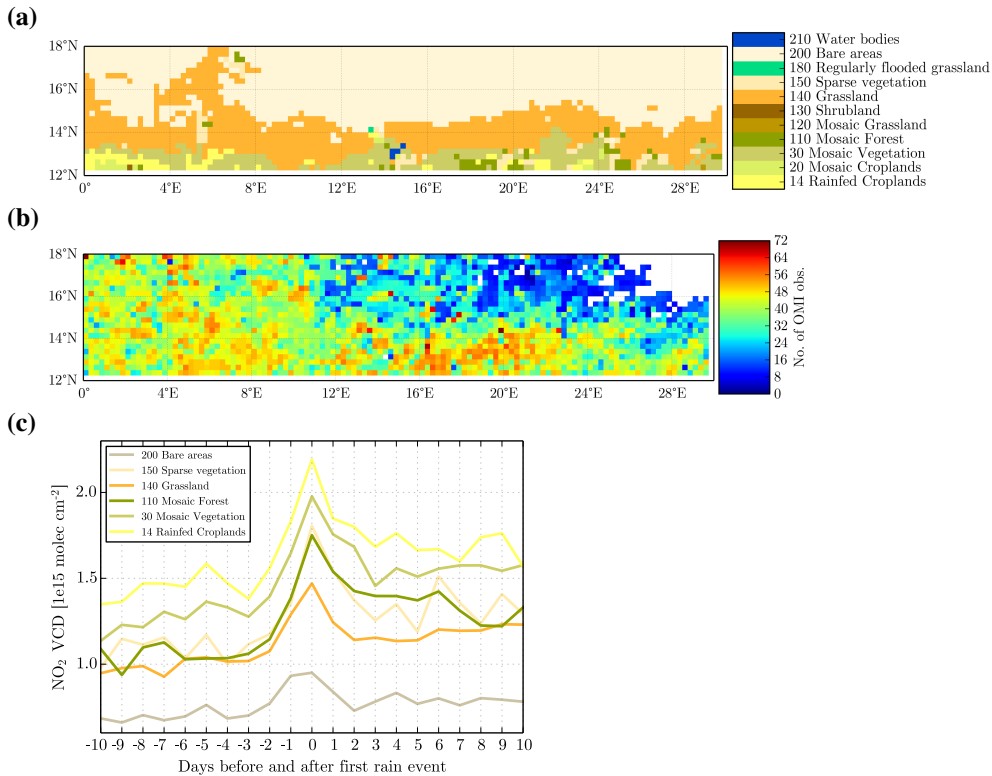

**Figure 6. (a)** ESA GLOBCOVER land cover classification for the Sahel down-scaled to 0.25° x 0.25 ° resolution with the corresponding (official) identification number, short name and color information for each class. **(b)** Spatial location and number of OMI observations for the reference case analysis (AMJ, 2007–2010). **(c)** Rain-triggered $NO_2$ enhancements from OMI for the Sahel region separated by the dominant land cover types.

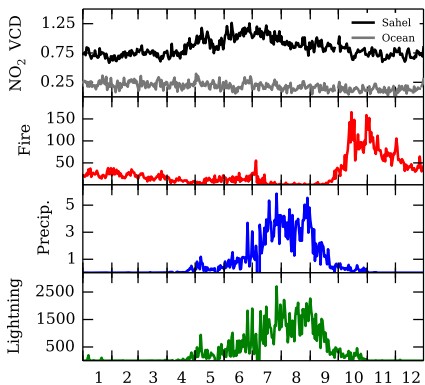

**Figure 7.** Daily time series for the Sahel region (0-30° W, 12-18° N) averaged for the years 2007, 2008, 2009 and 2010. The first row of each panel shows mean $NO_2$ VCDs from OMI in $\text{molecules cm}^{-2}$ (black) and a clean ocean reference (grey, 130-150°W, 12-18°N). The second row shows the number of active fire counts in the Sahel from MODIS. The third row shows average precipitation from the TMPA/TRMM product in mm. The fourth row shows the number of lightning strikes detected by WWLLN.

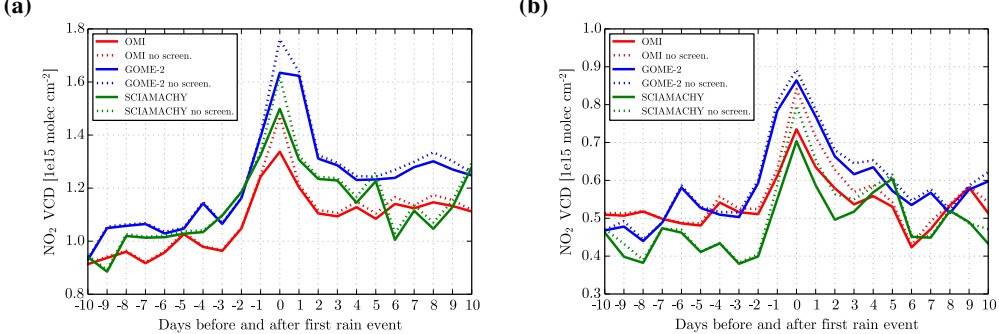

**Figure 8.** Effect of lightning screening on the response of $NO_2$ VCDs around the first day of rainfall after a prolonged dry spell. **(a)** $NO_2$ VCDs for the Sahel region from SCIAMACHY, GOME-2 and OMI without lightning screening (dashed lines) and with lightning screening (solid lines). **(b)** The corresponding results for Central Australia (15–30° W, 2–10° S).

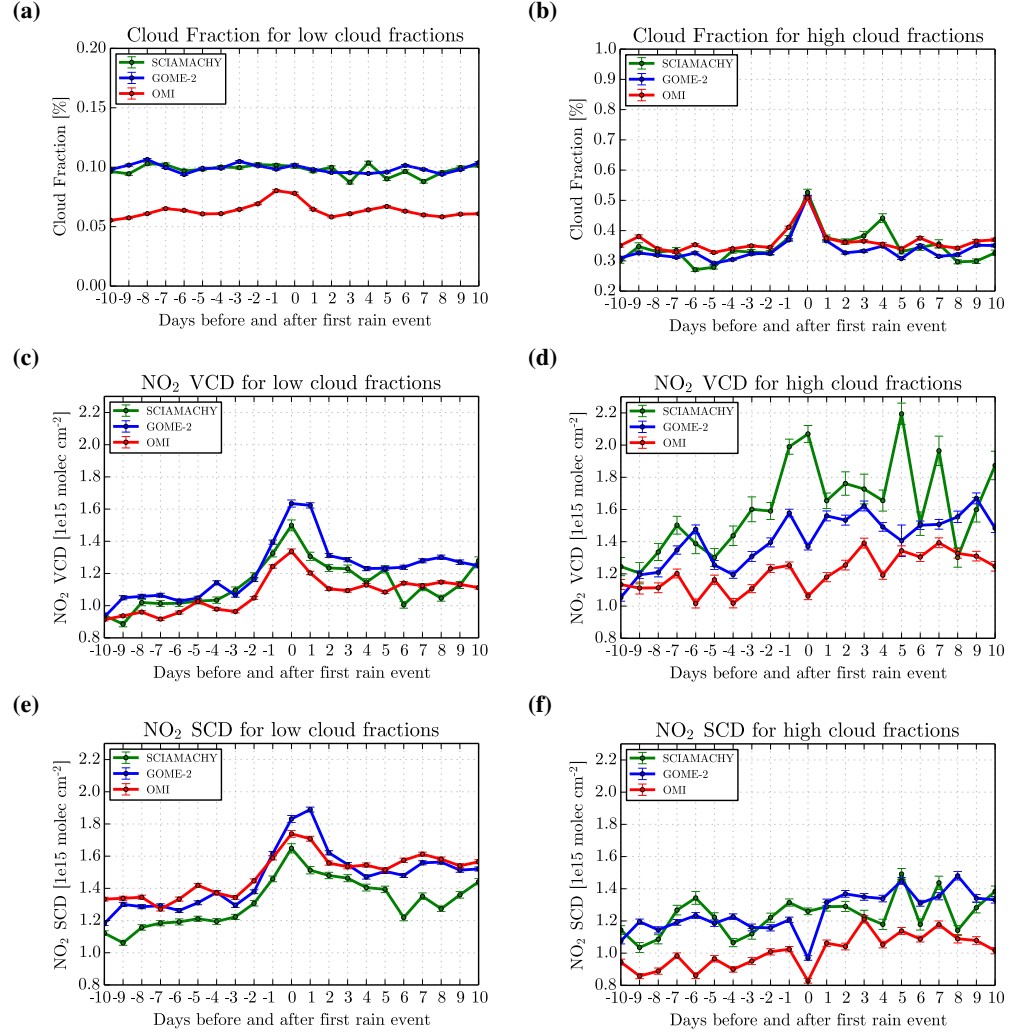

**Figure 9.** Investigation of cloud effects on the retrieved $NO_2$ SCDs and VCDs. **(a)** Mean cloud fraction of the three satellite instruments for the reference case (cloud fraction <20%). **(b)** Mean cloud fraction of the three satellite instruments but considering only observations with cloud fraction > 20%. **(c)** $NO_2$ VCDs for the reference case. **(d)** $NO_2$ VCDs only considering observations with high cloud cover (cloud fraction > 20%). **(e)** $NO_2$ SCDs for the reference case. **(f)** $NO_2$ SCDs only considering observations with high cloud cover (cloud fraction > 20%).

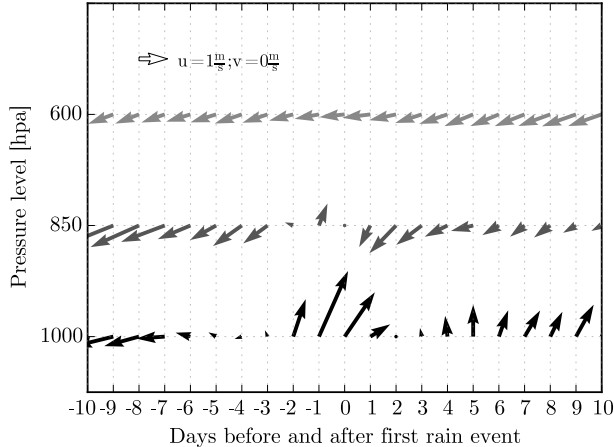

**Figure 10.** Mean ECMWF wind vectors at three different pressure levels. At the surface, a strong south-westerly wind is blowing the two days before the first rain event in the Sahel region followed by northerly winds. At 600 hPa winds are constantly from the north-west.

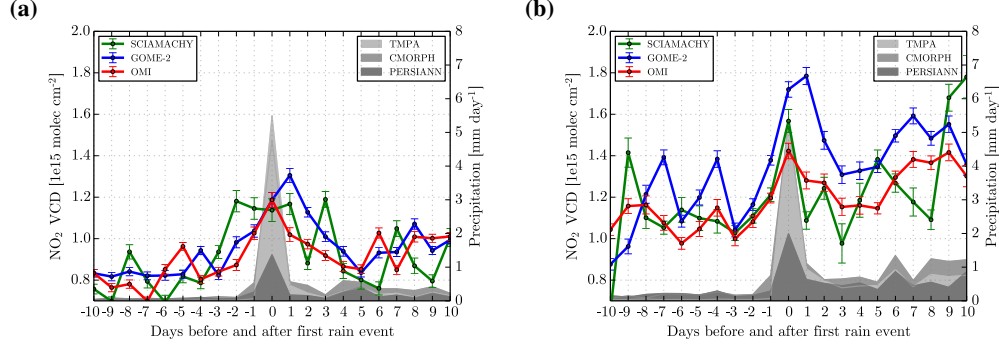

**Figure 11.** Same as Fig. 5d, but filtered for **(a)** northerly and **(b)** southerly winds. The filter criterion is fulfilled, if the wind direction at 600, 850, and 1000 hPa is North or South, respectively.

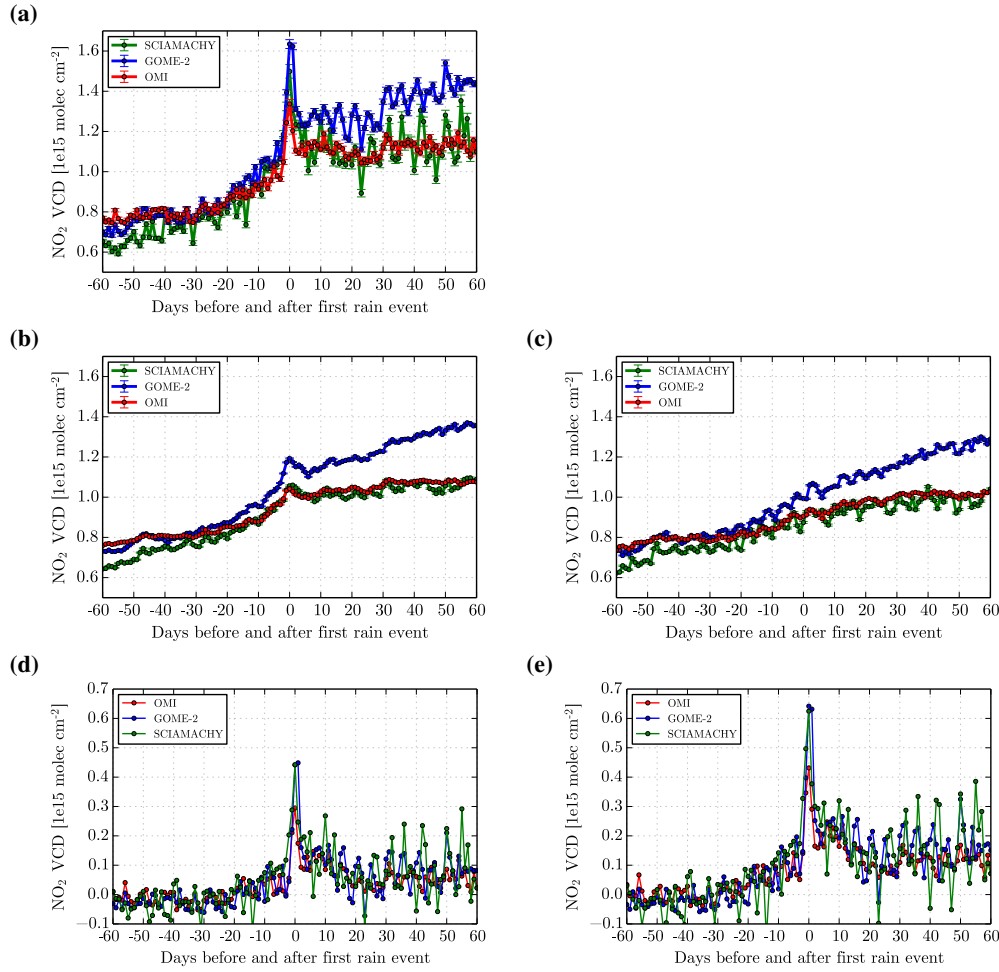

**Figure 12.** Analysis of $NO_2$ VCDs 60 days before and after the first day of rainfall in the April-May-June period for the Sahel region. **(a)** $NO_2$ VCDs from OMI, GOME-2 and SCIAMACHY. **(b)** Latitudinal averaged $NO_2$ VCDs corresponding to the reference (for details see text). **(c)** Latitudinal averaged $NO_2$ VCDs corresponding to the reference considering an additional buffer of 10 pixels around the actual triggered pixel to avoid influence of enhanced $NO_2$ VCDs in the vicinity of the precipitation events. **(d)** Background corrected $NO_2$ VCDs: panel (a) - panel (b) **(e)** Background (with buffer) corrected $NO_2$ VCDs: panel (a) - panel (c)

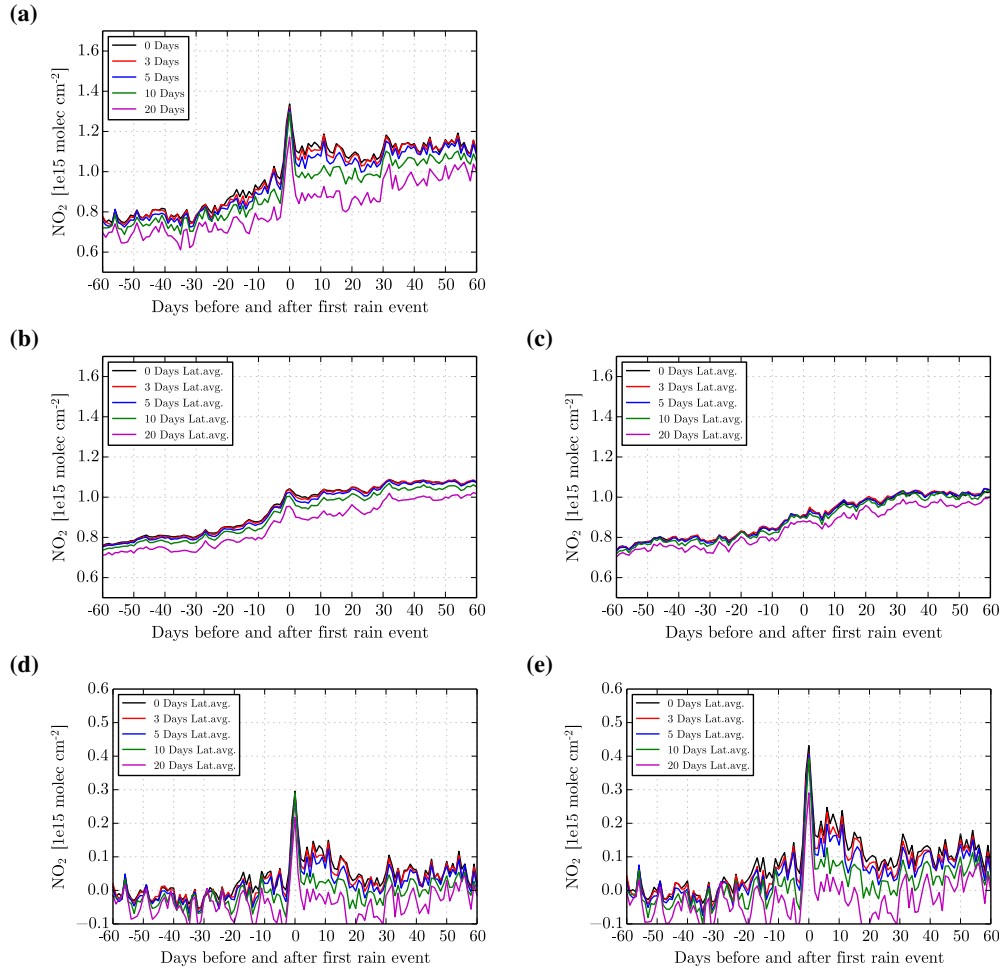

**Figure 13.** Investigation of the effect of different periods of dry days following Day0. **(a)** Reference case analysis for OMI $NO_2$ VCDs filtered for time series experiencing 0, 3, 5, 10 and 20 days of drought after Day0. **(b)** Background time series of $NO_2$ VCDs without buffer screening as presented in Fig. 12b for the corresponding time series experiencing 0, 3, 5, 10 and 20 days of drought after Day0. **(c)** The corresponding background time series of $NO_2$ VCDs with buffer screening as presented in Fig. 12c. **(d)** Differences between panels (a) and (b). **(e)** Differences between panels (a) and (c).

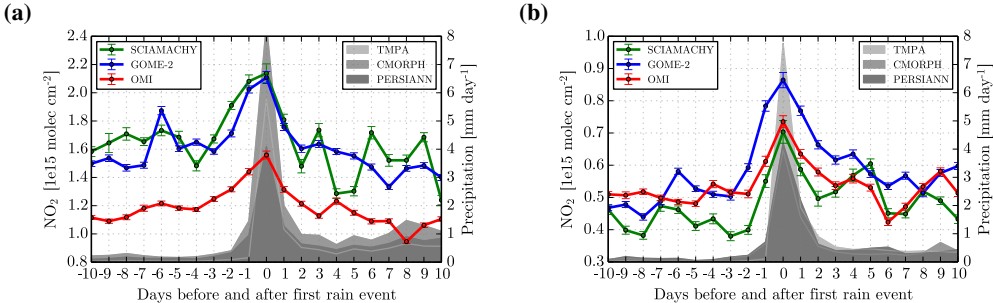

**Figure 14. (a)** $NO_2$ VCDs from SCIAMACHY, GOME-2 and OMI for South West Africa in September-October-November around the first day of rainfall in this period. Precipitation is represented by the grey shaded areas. **(b)** Results for Central Australia for the complete years 2007–2010.

## Appendix A:  Reference Case for different precipitation products as trigger

In this section, $NO_2$ VCDs are shown (akin to Fig. 5.e) from OMI (**A1**), GOME-2 (**A2**), SCIAMACHY (**A3**) around the first day of rainfall for different precipitation products as trigger for the precipitation threshold of 2 mm for the Sahel region. The left panels represent the full time series of 60 days before and after Day0, the right panels show a zoom-in for the 10 days before and after.

### A1   TMPA

### A2   CMORPH

### A3   PERSIANN

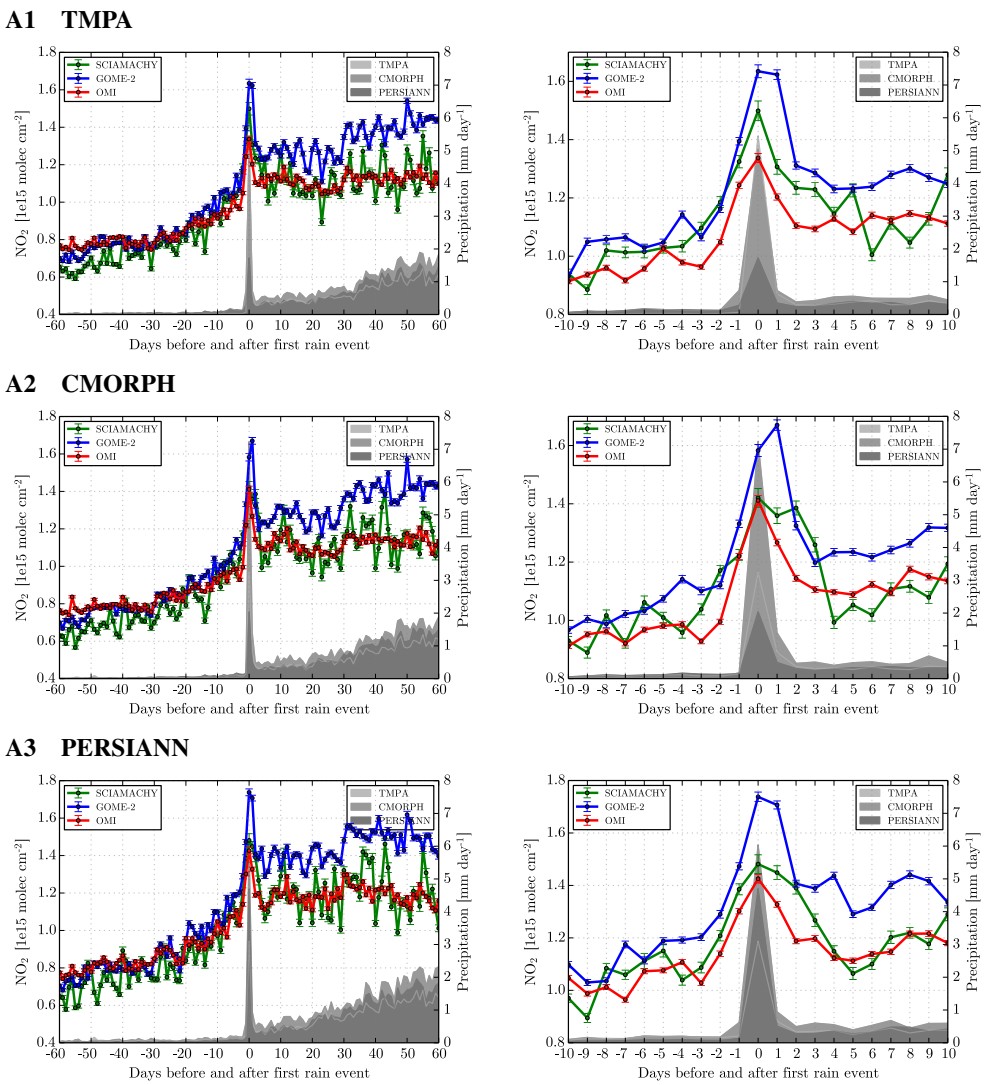

## Appendix B: Different drought lengths for the reference case

In this section, $NO_2$ VCDs are shown from OMI (**B1**), GOME-2 (**B2**), SCIAMACHY (**B3**) around the first day of rainfall for different preceding drought periods for the Sahel region with a precipitation threshold of 2 mm. For better intercomparison, the latitudinal background correction with buffer as described in section 4.5 is applied to each time series individually. The left panels represent the full time series of 60 days before and after Day0, the right panels show a zoom-in for the 10 days before and after.

**B1    OMI**

**B2    GOME-2**

**B3    SCIAMACHY**

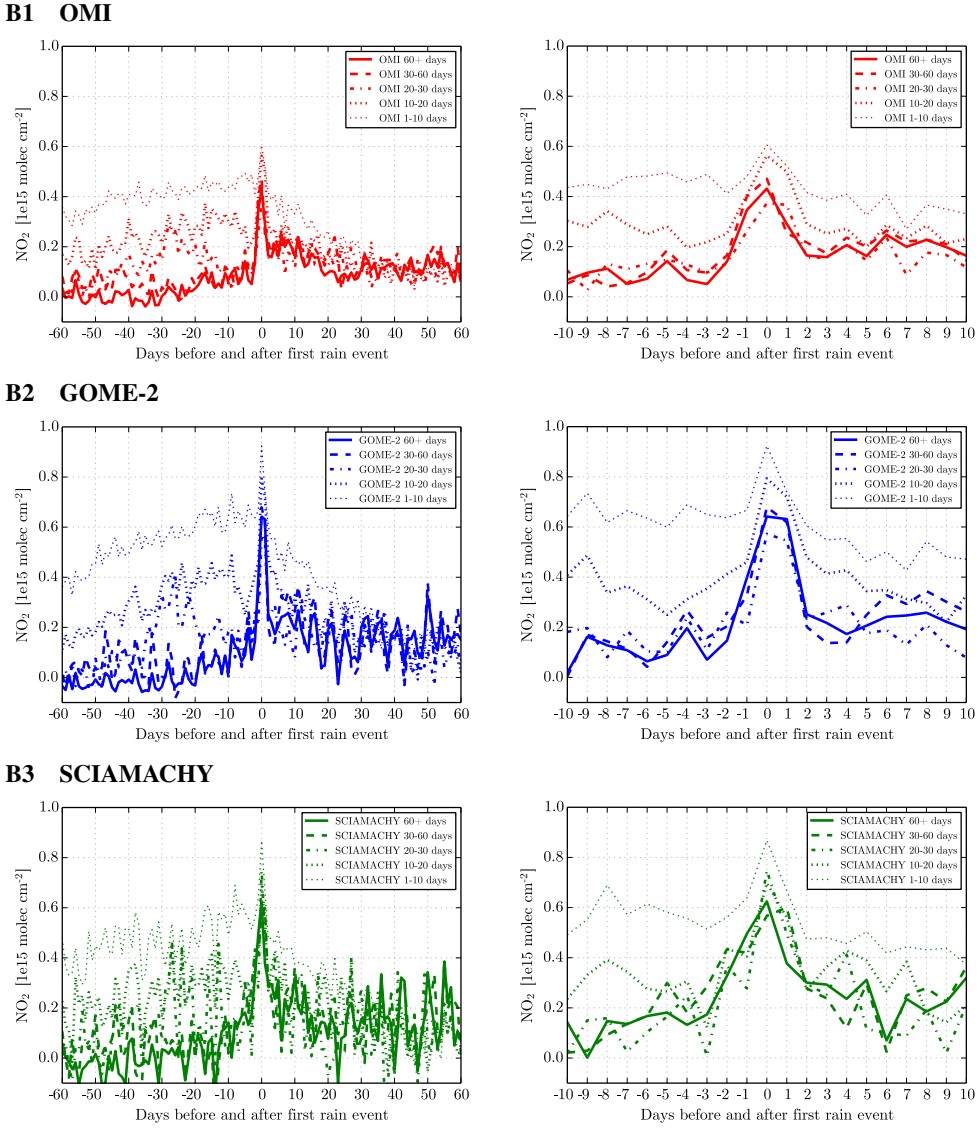

## Appendix C:  Different precipitation thresholds on Day0 for the reference case

In this section, $NO_2$ VCDs are shown from OMI (**C1**), GOME-2 (**C2**), SCIAMACHY (**C3**) around the first day of rainfall for the Sahel region for a drought period of at least 60 days. The results are separated for different intervals of the precipitation threshold on Day0. A drought day is defined by precipitation < 0.1 mm per day. For better intercomparison, the latitudinal background correction with buffer as described in section 4.5 is applied to each time series individually. The left panels represent the full time series of 60 days before and after Day0, the right panels show a zoom-in for the 10 days before and after.

### C1   OMI

### C2   GOME-2

### C3   SCIAMACHY

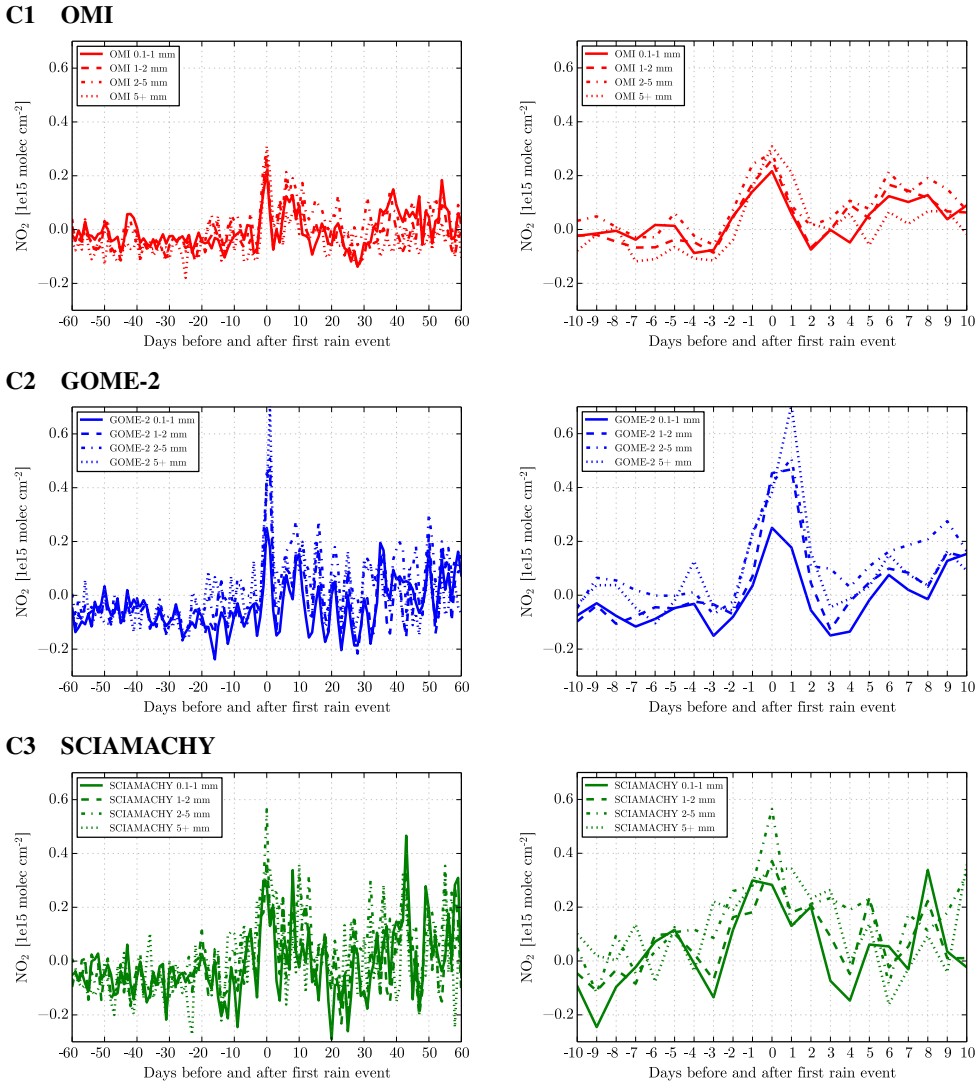

**Appendix D: Global Maps for other satellite instruments (60 days of drought, 2 mm precipitation threshold)**

In this section, absolute differences in NO$_2$ VCDs compared to Days -10 to -2 from GOME-2 (**D1**) and SCIAMACHY (**D2**) on Day0 (first day of rainfall) are depicted which were computed similarly as for OMI in Fig. 3.d for OMI. The data was screened for significant changes and pixels containing less than 20 measurements on Day0 (or less then 50 measurements from Day-10 to Day-2).

### D1  GOME-2

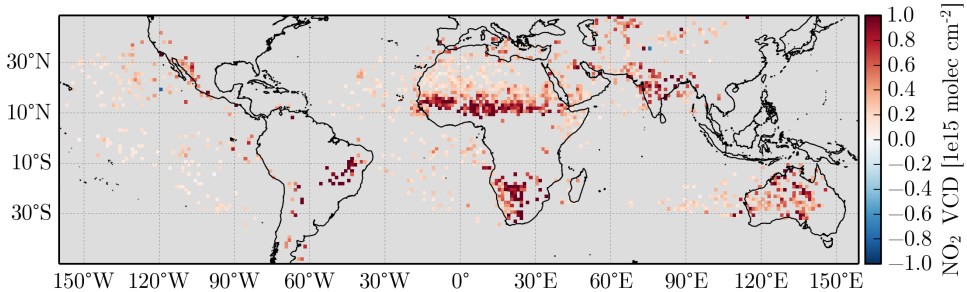

### D2  SCIAMACHY

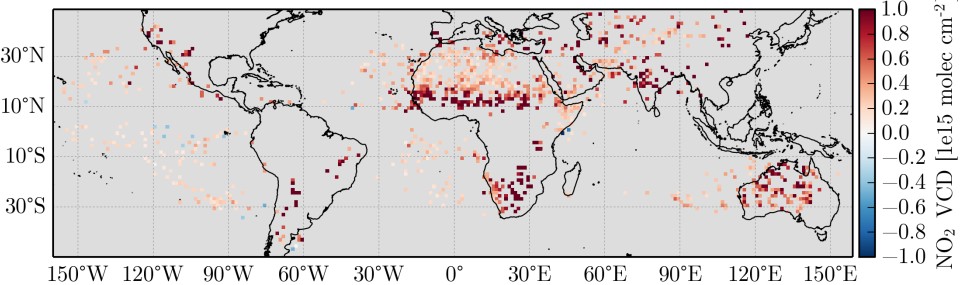

## Appendix E:  Impact of a-priori precipitation product

In this section, the impact of the precipitation product on the derived soil $NO_x$ emissions is investigated. Figures **E1** and **E2** depict the $NO_2$ enhancement on Day0 as in Fig. 3d, but based on CMORPH and PERSIANN data, respectively. While the absolute values differ for the Sahel, the final emission estimates for pulsed emissions are quite similar (see Table **E3**), as the choice of the precipitation data affects the background correction as well.

### E1  CMORPH

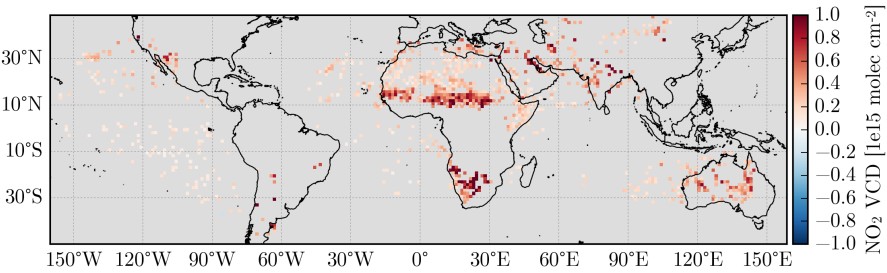

### E2  PERSIANN

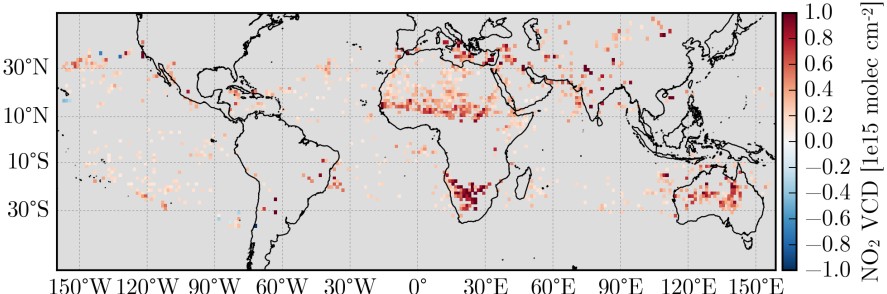

### E3  Derived Fluxes

|  | **TRMM** | CMORPH | PERSIANN |
|---|---|---|---|
| ⌀ Day0 enhanc. [ngNm$^{-2}$s$^{-1}$] | **6.95** | 7.75 | 6.62 |
| max. Day0 enhanc. [ngNm$^{-2}$s$^{-1}$] | **64.61** | 64.61 | 64.61 |
| Day1-14 enhanc. [ngNm$^{-2}$s$^{-1}$] | **3.39** | 3.07 | 2.42 |
| Background (soil) [ngNm$^{-2}$s$^{-1}$] | **2.75** | 3.39 | 4.36 |
| Background (total) [ngNm$^{-2}$s$^{-1}$] | **14.54** | 15.18 | 16.15 |

## Appendix F: Analysis for different soil temperatures

In this section, NO$_2$ VCDs are shown (akin to Fig. 5.e) from OMI around the first day of rainfall for different soil temperatures on Day0 for the Sahel region (**left panel**). The **right panel** depicts daily time series of soil temperature (from 12UTC ECMWF data) for the Sahel region (0-30$^\circ$W, 12-18$^\circ$N) averaged for the years 2007, 2008, 2009 and 2010.

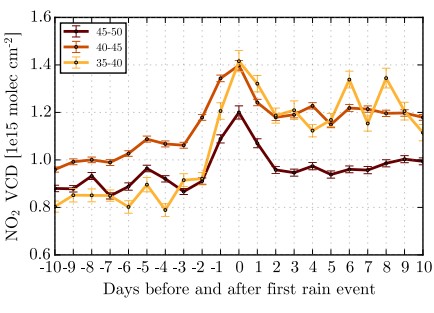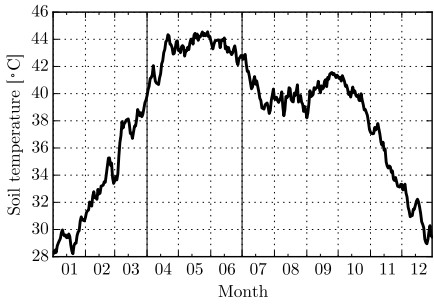