# Peer review of "Multi-satellite sensor study on precipitation-induced emission pulses of $NO_x$ from soils in semi-arid ecosystems"

_Atmospheric Chemistry and Physics, 2016_

## Referee Comment (RC1) · Anonymous Referee #1 · 9 Mar 2016

Review on manuscript acp-2016-93. Multi-satellite sensor study on precipitation-induced emission pulses of NOx from soils in semi-arid ecosystems By J. Zörner, M.J.M. Penning de Vries , S. Beirle , H. Sihler , P.R. Veres, J. Williams, and T. Wagner.

The topic of the paper is about a top down approach to determine soil NOx emissions provoked by first rains after a long period of drought. Satellite data from three different instruments are used to retrieve NO2 Vertical Column Densities, as well as other parameters necessary to interpret soil NOx emissions variability. The study is global,, but focuses on semi arid regions, and specifically on the Sahel region. This study shows that increases in NO2 VCDs are explained by soil NOx emissions at the beginning of the rain season. Corresponding fluxes are calculated and are in the range of literature

values.

General comments

The paper is totally within the scope of ACP. The idea that pulsing emissions occur on very dry soil after the first rain is not new, but it is presented and demonstrated in a very comprehensive way including three different satellite instruments. Biases and possible misinterpretations of satellite data are ruled out and the conclusions are convincing.

The results are correctly presented; the figures illustrate the results in a clear way. The paper is well written, regardless of some minor corrections. I recommend this paper to be published in ACP, after minor corrections and improvements.

The principal missing points concern: - a calculation of the budget at the seasonal scale in TgN for the Sahel. The evaluation of NO fluxes from soils is a good point, but it would have been interesting to know the overall budget at the regional scale. All the necessary information is available (fluxes, area) for this calculation. If this calculation cannot be provided please explain why precisely.

- References on soil NOx emissions are a bit old, except Hudman et al. (2012). New references could be included, see suggestions below.

- Too little explanations are given on processes responsible for soil NOx emissions. References are suggested below.

- The Western Sahel is not included in the study, the reasons why are not clear and must be detailed.

Specific comments

Abstract. The abstract gives a clear idea of what is presented in the paper.

Introduction Line 7, page 2: NOx is also removed by NO2 deposition on vegetated surfaces.

Line 23 page 2: explanations of nitrification and denitrification processes are a bit confused. Please refer to Pilegaard et al., Phil Trans. R.. Soc., 2013.

Line 31 page 2: you should also mention HONO emissions from semi arid soils, see Oswald et al., Science 341, 1233, 2013. "N-fixing microbial species occur": not clear enough

Line 6 page 3: Soils emission depend also on pH, N content (not only N input).

Line 8 page 3: In the Sahel, the presence of cattle is an important provider of organic fertilization. This should be mentioned. See for example Delon et al. (2010), already referenced in your paper.

Line 11 page 3: Add some recent publications of pulsing. Such as Kim et al., Biogeosciences, 9, 2459–2483, 2012, Wang et al., Volume 6(8), Article 133, Ecosphere, 2015.

Line 6 page 4: Section 3 is mentioned, but sections 1 and 2 should be mentioned first.

Line 7 page 4: "This approach. . ." should be "In section 4, this approach. . ."

Line 10 page 4: precise which governing parameters you refer to.

Line 11 page 4: "also" is not at the right place in the sentence.

Line 19 to 26 page 4: the way of presenting the different points ((i) (ii) (ii)) is not easy to read. Making proper sentences would be more readable.

Line 19 page 6: "Relative fluxes": relative to what?

Line 27 page 6: reformulate sentence beginning with "The data sources. . ."

Line 13 page 7: Sentence beginning with "The Moderate. . ." is difficult to understand. Please reformulate

Line 17 page 7: Specify "others". Specify the time period and the time resolution used.

Line 5 page 8: the time period is précised here, it should be precised earlier.

Line 16 page 8: "in the Sahel and shorter" should be "in the Sahel to shorter periods"

Line 20 page 8: precise that background level is precipitations < 2 mm during 60 days.

Line 3 page 9: Mention that fig 3 will be described below in the Results paragraph.

Line 15 page 9: Ad "the" between "Although" and "best". This sentence is confusing, in the sense that you write that analysis cannot be not in the Tropics, Northern America, Europe, South Asia? Please explain.

Line 23-24 page 9: Sentences need to be reformulated.

Line 2 page 10: Transports are mentioned to explain NO2 VCDs enhancements. Neither transports nor industries and traffic were mentioned in the introduction as possible sources. Why should transport explain enhanced emissions at the first day of rain?

Line 7 page 10: the dry season in the Sahel lasts nearly 8 months. Was the month of July tested as part of the months when the first day of rain occur? Sometimes when the wet season is late, the first day of rain occurs only in July. See Lebel et al., Journal of Hydrology 375 (2009) 52–64.

Line 14 page 10: add "next" after "gradually with the start of the..."

Line 19 page 10: Why did you exclude the western Sahel of the studied zone?

Lines 22-25: this paragraph should be in the methodology section.

Line 24 page 10: You mean that for a period less than 2 months, the N enrichment is not sufficient? Can you give references for that?

Line 11 page 11: smaller instead of smallest.

Line 25 page 11: the expression "dry phases" is not understandable here. It is only understandable once reading the following sections. May be a word or two to explain could be useful.

Lines 27-30, page 11: do you think the enhancement of HCHO the day before Day0 has a link with air moisture? What are the processes that may be involved?

Lines 6-7 page 12: industrial activities and strongly fertilized agriculture are hardly found in the Sahel even in the Southern part. You mean may be the southern part of West Africa?

Line 9 page 12: low nitrogen input and nitrogen content. The role of cattle should be developed in this paragraph. Mineral fertilizers are not widely used in the Sahel, while organic fertilization plays a non negligible role in sNOx emissions. See for example Schlecht and Hiernaux, Nutrient Cycling in Agroecosystems 70: 303–319, 2004.

Line 20 page 13: a short conclusion of possible under or overestimation of cloud effect on NO2 VCDs should be useful.

Line 9 page 14: ",thus," may be removed from the sentence.

Line 22 page 14: "at" instead of "on" the same latitude.

Line 16 page 15: "largest" instead of "larger".

Line 3 page 16: specify "Eastern" Sahel, because the analysis has not been made in Western Sahel.

Line 5 page 18: As indicated in Aghedo et al., Atmos. Chem. Phys., 7, 1193–1212, 2007, anthropogenic pollution is not likely to reach sahelian latitudes.

An important added value could be brought here to this paper. As mentioned in the general comments, the budget over the whole studied area (i.e. Eastern Sahel) could be calculated in TgN for the studied period.

Conclusions Lines 7-13: difficult to follow with this a) to e) way of presenting the ideas. Proper sentences would be more readable.

Line 15 page 18: "maximum amount of precipitation"? do you mean the 2 mm thresh-

old? In that case it is the minimum amount.

Line 17 page 18: "shown" instead of "showed".

Line 20 page 18: again see Oswald et al., 2013, where laboratory measurements made on semi arid soils are presented.

Technical corrections Line 11 page 3: "lab" should be laboratory

Line 13 page 6: Upper case to begin the paragraph is needed.

e.g. throughout the text should be in italics.

————————————————————

---

## Referee Comment (RC2) · Anonymous Referee #2 · 25 Mar 2016

General comments

The authors have conducted an in depth analysis of soil NOx emission responses to rain events preceded by a dry period. This is a significant analysis in that it was done at the global scale with high resolution, both spatially and temporally. The authors also conducted an in depth and satisfying evaluation of the factors that can lead to errors in using a top-down approach for estimating soil NOx fluxes, including lightning, fire, meteorology, transport, cloud cover and AMF corrections etc.

While I have few criticisms of the analysis itself I do have trouble understanding how this analysis significantly enhances our understanding of global soil NOx emissions. Synthesis and interpretation of the results were unsatisfying. For example, there is

a confusing amount of time spent on correcting for background emissions following a pulse event. What is the purpose of this? Is it to advance modeling efforts? If so, there needs to be some direct connection of the results to modeling or at least a proposed way in which to use these results to inform modeling.

One of the unique and valuable aspects of the analysis is spatial resolution at the global scale however the authors focus much of the paper on a single event in the Sahel, a phenomenon many other papers have already focused on. The conclusion section does not even mention the global analysis except to say that it was done and confirm that semi-arid regions of the world are likely to have soil NOx pulses. The significance of these pulse emissions at the global scale should be quantified and more clearly presented in order to show their significance within the global NOx budget and how it has advanced our understanding of the global NOx budget.

Also, within the Sahel analysis, it is again not clear how the results enhance our understanding of soil NOx pulses in the Sahel beyond which we already know.

Overall, I have few criticisms of the analysis itself, just of the interpretation and presentation of the data.

Specific comments

On page 3 line 19, it is stated that soil NOx pulses are only enhanced for 1-3 days post precipitation, however other studies have shown pulses to last much longer, up to 25 days. (See Oikawa et al. Unusually high soil nitrogen oxide emissions influence air quality in a high temperature agricultural region, Nature Communications 2015)

Pg 6 line 17, Authors state only minor effects resulting from uncertainty in precipitation events across 3 data products. However it would be preferable to quantify that uncertainty or at least state the maximum and average amount of deviation there is across the data products for different regions. Appendix A shows only 1 example.

Pg.7 line 20. Please provide at least a discussion of the error associated with land

cover data sets.

Line 19 Pg 11–The authors refer to error caused by AMF several times however never indicate any quantification of that error, or suggest references that have investigated error in data products such as in OMI. After filtering for cloud cover, for example, what amount of error is expected to remain?

On pgs 14-15 there is a large discussion of whether enhanced VCD's are the result of precipitation on Day 0 vs precipitation generally being enhanced following that first rain event, aka seasonal changes. For example, the authors state "However, it still needs to be clarified whether the enhanced NO2 VCDs after Day0 are induced by the initial precipitation on Day0 or by continuous precipitation during the following days." But it is not clear to the reader why this distinction is important.

---

## Author Comment (AC1) · 3 Jun 2016

**Multi-satellite sensor study on precipitation-induced emission pulses of NOx from soils in semi-arid ecosystems by J. Zörner et al.**

**Reply to anonymous referee #1**

We would like to thank reviewer #1 for the very constructive and encouraging comments and suggestions. Our replies as well as the changes made to the manuscript are provided below.

Blue: Reviewer comment
Black: Author's reply
Red: Modified text in manuscript

**General comments:**

(1) a calculation of the budget at the seasonal scale in TgN for the Sahel. The evaluation of NO fluxes from soils is a good point, but it would have been interesting to know the overall budget at the regional scale. All the necessary information is available (fluxes, area) for this calculation. If this calculation cannot be provided please explain why precisely.

Such a calculation certainly helps to put the total amount of emitted N from soils in a perspective to other N sources and the total N budget. Therefore, we included an additional subsection to the discussion part and updated the other parts of the manuscript correspondingly. Furthermore, we added average time series for $NO_2$ VCDs, precipitation, fire and lightning to the manuscript to further illustrate variations in background $NO_2$ VCDs as well as potential causes.

[revised manuscript text omitted]

The introductory section is updated accordingly. Proposed changes to the manuscript are provided in the answer to question #10.
* * *
(3) Too little explanations are given on processes responsible for soil NOx emissions. References are suggested below.

Proposed changes to the manuscript can be found below in the document (see reply to question #6).
* * *
(4) The Western Sahel is not included in the study, the reasons why are not clear and must be detailed.

We agree with the Referee that a region West of our main study region also shows enhanced $NO_2$ columns in response to the first rain (Fig. 4a). Thus, we added the argumentation on why we focus on the Central and Eastern part of the Sahel to the beginning of section 4.2.

We restrict our detailed analysis to the Central and Eastern part of the Sahel region (0-30° E, 12-18° N), similar as in previous studies (i.e., Jaeglé et al. 2004, Hudman et al. 2012). The western part of the Sahel shows a slightly weaker $NO_2$ response to rain pulses, which might be related to different inter-annual variability patterns and seasonal cycles of precipitation regimes (Lebel and Ali, 2009).

**Specific comments:**
* * *
(5) Introduction Line 7, page 2: NOx is also removed by NO2 deposition on vegetated surfaces.

We added the following reference and changed the manuscript accordingly:
Ganzeveld LN, Lelieveld J, Dentener FJ, Krol MC, Bouwman AJ, Roelofs G-J. 2002 Global soil-biogenic NOx emissions and the role of canopy processes. J. Geophys. Res. 107, ACH9-1–ACH9-17, doi:10.1029/2001JD001289

*Line 10, Page 2:* Furthermore, $NO_x$ is also removed by $NO_2$ deposition on vegetated surfaces (Ganzeveld et al. 2002).
* * *
(6) Line 23 page 2: explanations of nitrification and denitrification processes are a bit confused. Please refer to Pilegaard et al., Phil Trans. R.. Soc., 2013.

We updated this paragraph accordingly and also added corresponding references.

Emissions of $NO_x$ from natural and anthropogenically influenced soils are mainly driven by microbial activity within the top soil layer and associated chemical reactions (Conrad, 1996). Primarily, two important groups of micro-organisms, nitrifiers and denitrifiers, are involved in processes related to the turnover of nutrients in the soil (Pilegaard et al., 2013, Behrendt et al. 2015). They are directly responsible for the corresponding processes of: (i) nitrification, the biological oxidation of nitrogen compounds, typically the oxidation of soil ammonium ($NH_4^+$) to nitrate ($NO_3^-$) and (ii) denitrification, the reduction of nitrate by microbes to gaseous products, i.e. $N_2O$ and finally $N_2$. NO is a gaseous by-product of both processes and once released reacts with ambient $O_3$, to form $NO_2$ and oxygen ($O_2$) within minutes.
* * *
(7) Line 31 page 2: you should also mention HONO emissions from semi arid soils, see Oswald et al., Science 341, 1233, 2013. "N-fixing microbial species occur": not clear enough

We added HONO emissions to the beginning of the paragraph. Furthermore, we concretized the statement on enhanced nitrogen gas emissions under the presence of N-fixing organisms.

Findings from Oswald et al. (2013) suggest that gaseous nitrous acid (HONO), which is rapidly photolyzed

to NO, is also emitted from soils.

Emissions of nitrogen-containing gases, such as NO, $N_2$ and $N_2O$ increase dramatically in soils with enhanced nitrogen availability due to the presence of N-fixing microbial species and plants (Virginia et al., 1982, van Groenigen et al., 2015).
* * *
(8) Line 6 page 3: Soils emission depend also on pH, N content (not only N input).

We added these two properties and updated the paragraph accordingly.

Soil emissions of trace gases depend on a wide range of ambient environmental conditions such as soil type, soil moisture, temperature, pH-Value and nitrogen content (Conrad, 1996; Ludwig et al., 2001; Meixner and Yang, 2006; Oswald et al., 2013).
* * *
(9) Line 8 page 3: In the Sahel, the presence of cattle is an important provider of organic fertilization. This should be mentioned. See for example Delon et al. (2010), already referenced in your paper.

We agree with the referee that this type of fertilization should be mentioned. Therefore, we inserted the following statement:

In remote regions like the Sahel, where synthetic fertilizer is limited, manure plays a prominent role in the fertilization of agricultural fields and can contribute significantly to the input of organic nitrogen into the soil (Schlecht and Hiernaux, 2004; Delon et al., 2010).
* * *
(10) Line 11 page 3: Add some recent publications of pulsing. Such as Kim et al., Biogeosciences, 9, 2459–2483, 2012, Wang et al., Volume 6(8), Article 133, Ecosphere, 2015.

We like to thank the referee for the suggested literature on $NO_x$ emissions from soils and updated the paragraph accordingly.
* * *
(11) Line 6 page 4: Section 3 is mentioned, but sections 1 and 2 should be mentioned first.

We inserted the following note at the beginning of the paragraph:

The paper is organized as follows: in section 2, all data products used within this study are presented.
* * *
(12) Line 7 page 4: "This approach..." should be "In section 4, this approach..."

Done. The new sentence reads:

In section 4, this approach is then applied to areas with different spatial extents.
* * *
(13) Line 10 page 4: precise which governing parameters you refer to.

Done. The new sentence reads:

In a second step, we focus on Africa and the Sahel region, in specific, and separate the analysis for different seasons. For this region, we investigate fundamental relationships between soil emissions and some of their governing parameters, i.e. soil moisture content, temperature, air humidity.
* * *
(14) Line 11 page 4: "also" is not at the right place in the sentence.

The updated sentence now reads:

Within this analysis possible interferences from other parameters are also investigated, and detailed sensitivity studies are conducted.
* * *
(15) Line 19 to 26 page 4: the way of presenting the different points ((i) (ii) (ii)) is not easy to read. Making proper sentences would be more readable.

We updated this paragraph as suggested. The edited lines now read:

Tropospheric VCDs are usually derived in a multi step process (e.g. Boersma et al., 2004, 2007; De Smedt et al., 2008, 2012). First, total slant column densities (SCDs) are retrieved, i.e. the integrated concentrations along the effective light path, by fitting the measured spectrum with a model taking into account all other absorbers in the atmosphere. Second, tropospheric SCDs are derived by subtracting the stratospheric column ($NO_2$) or a latitude-dependent bias estimated over the Pacific (HCHO). Third, the tropospheric SCDs are then translated to tropospheric VCDs.
* * *
(16) Line 19 page 6: "Relative fluxes": relative to what?

We deleted the word "relative" and exchanged it with are more concise description of the relation.

The processes of nitrification and denitrification, which govern sNOx fluxes, are closely related to the soil water content (Meixner and Yang, 2006).
* * *
(17) Line 27 page 6: reformulate sentence beginning with "The data sources..."

The updated sentence now reads:

The data sources include active (scatterometer) and passive (radiometer) microwave observations acquired preferentially in the low-frequency microwave range.
* * *
(18) Line 13 page 7: Sentence beginning with "The Moderate..." is difficult to understand. Please reformulate

Done. The updated sentence now reads:

The MODIS global monthly fire location product MCD14ML (Terra and Aqua combined, Giglio et al., 2006) is used to filter out locations affected by fires.
* * *
(19) Line 17 page 7: Specify "others". Specify the time period and the time resolution used.

Done. The updated paragraph now reads:

In order to understand the prevailing meteorology and filter for special circumstances in the Sahel region, modelled data of air temperature, pressure, humidity as well as wind fields are taken from the ECMWF ERA-Interim analysis (Dee et al., 2011). The model data is acquired at a spatial resolution of 0.25° and a temporal resolution of 6 hours over the period from 2007 to 2010. The data is publicly available via http://apps.ecmwf.int/datasets/.
* * *
(20) Line 5 page 8: the time period is precised here, it should be precised earlier.

We added this information in the introduction.

(ii) high spatial resolution, which is both achieved by expanding the time span of the study to several years (2007 to 2010) enabling an investigation of single grid pixels of 0.25° with reasonable statistics.
* * *
(21) Line 16 page 8: "in the Sahel and shorter" should be "in the Sahel to shorter periods"

The edited sentence now reads:

However, the length of drought phases are quite different for semi-arid areas in the world, varying from very long (several months in winter) in the Sahel to shorter periods (several weeks to months in summer) in South West Africa.

(22) Line 20 page 8: precise that background level is precipitations < 2 mm during 60 days.

We realize the concern of the referee. However, we think this issue is already sufficiently explained by the sentence before.

(23) Line 3 page 9: Mention that fig 3 will be described below in the Results paragraph.

We thank the referee for this note as it reveals a typo in the manuscript. Figure 3 should not be mentioned in the methodology. Instead, we refer to figure 2 in the paragraph before. We updated the corresponding part to:

In the example shown in Fig. 2 a 0.25° x 0.25° pixel is chosen which provides a complete $NO_2$ time series over 10 days.

(24) Line 15 page 9: Ad "the" between "Although" and "best". This sentence is confusing, in the sense that you write that analysis cannot be not in the Tropics, Northern America, Europe, South Asia? Please explain.

We changed this sentence to improve the readability:

For most regions in the world enough data points are found for our analysis; exceptions are regions with no pronounced seasonality in rainfall (e.g., tropical rainforests, North America, Europe) and regions where rain occasionally falls during the dry season (Southeast Asia). Our algorithm is not optimized for those regions.

(25) Line 23-24 page 9: Sentences need to be reformulated.

We changed this sentence to:

The corresponding results for $NO_2$ VCDs observed by GOME-2 and SCIAMACHY are similar to Fig. 3d, but are more affected by noise due to poorer statistics (see appendix D).

(26) Line 2 page 10: Transports are mentioned to explain NO2 VCDs enhancements. Neither transports nor industries and traffic were mentioned in the introduction as possible sources. Why should transport explain enhanced emissions at the first day of rain?

We do not refer in this context to transportation related to traffic by ships or cars but to long-range transport of air masses. This potential influence was not mentioned in the introduction as this effect was not expected to be relevant for this study. However, for some grid pixels in proximity to coastal areas we find slightly enhanced $NO_2$ VCDs. This could be possibly explained by advection of polluted air synced with the moving precipitation system around the first day of rainfall (i.e. a change of wind direction and speed favouring local enhancements in $NO_2$ VCDs). For clarification we replace "transport" with "advection". The updated sentence now reads:

However, over the Mediterranean sea and in proximity to coastal regions over oceans small-scale enhancements in $NO_2$ VCDs are detectable which might be related to advection.

(27) Line 7 page 10: the dry season in the Sahel lasts nearly 8 months. Was the month of July tested as part of the months when the first day of rain occur? Sometimes when the wet season is late, the first day of rain occurs only in July. See Lebel et al., Journal of Hydrology 375 (2009) 52–64.

For the sensitivity studies as well as the emission calculation we only investigated the months April, May and June. Fig. 1 depicts the number of triggered rain events that fulfil a required threshold of 2 mm after 60 days of drought for individual pixels in the Sahel for the April-May-June period. It can be seen that within the 4 years (2007 to 2010) we investigate in this study, the first rain events of the wet season in the Sahel region all occur within these months.

[Figure]

Figure 1: Number of triggered rain events that fulfill a required threshold of 2 mm after 60 days of drought for individual pixels in the Sahel for April-May-June.

(28) Line 14 page 10: add "next" after "gradually with the start of the..."

The modified text now reads:
The subsequent dry season begins in October and ends gradually with the start of the next wet season in April/May/June (AMJ-period).

(29) Line 19 page 10: Why did you exclude the western Sahel of the studied zone?

We provide our answer and corresponding changes in the manuscript in question #4 of this document.

(30) Lines 22-25: this paragraph should be in the methodology section.

We moved this paragraph to the end of the methodology section.

(31) Line 24 page 10: You mean that for a period less than 2 months, the N enrichment is not sufficient? Can you give references for that?

We did not intend to convey that the N enrichment for a period less than two months is too short. It is meant that during drought N enrichment dominates over N depletion in the soil. In appendix B, we find

pulsing events for even short periods of about one week. However, the observed enhancements in $NO_2$ VCDs on Day0 are lower for such cases. For clarification we revised this paragraph as follows:

The drought period of about two months is chosen as we find the highest response in $NO_2$ with this setting. In appendix B, the impact of drought lengths on the derived soil emission pulses is investigated.
* * *
(32) Line 11 page 11: smaller instead of smallest.

Done.
* * *
(33) Line 25 page 11: the expression "dry phases" is not understandable here. It is only understandable once reading the following sections. May be a word or two to explain could be useful.

We updated the sentence to:

In sections 4.4 and 4.5, this important finding is studied more in detail by analyzing the $NO_2$ levels after Day0 depending on wind conditions and the precipitation on Day1 and beyond.
* * *
(34) Lines 27-30, page 11: do you think the enhancement of HCHO the day before Day0 has a link with air moisture? What are the processes that may be involved?

In our opinion, the presented HCHO data are dominated by noise and we do not attempt to interpret them.
* * *
(35) Lines 6-7 page 12: industrial activities and strongly fertilized agriculture are hardly found in the Sahel even in the Southern part. You mean may be the southern part of West Africa?

The original formulation indeed reflects the case for the more Western Part of the Sahel (which still overlaps with the Eastern region we investigate). Still, anthropogenic induced emissions are more frequent in the Southern part compared to the Northern part in the region which we investigate. Thus, we updated the sentence in the manuscript slightly:

Anthropogenic activity and related emissions such as domestic fires or fertilized fields are at a very low level, and originate mostly from the southern, more populated part of the Sahel (Delon et al., 2010).
* * *
(36) Line 9 page 12: low nitrogen input and nitrogen content. The role of cattle should be developed in this paragraph. Mineral fertilizers are not widely used in the Sahel, while organic fertilization plays a non negligible role in sNOx emissions. See for example Schlecht and Hiernaux, Nutrient Cycling in Agroecosystems 70: 303–319, 2004.

We would like to thank the referee for the this remark and the suggested reference. In this section, however, we think that manure is not central to our analysis. Thus, we propose an additional paragraph in comment #9 in the introduction which covers the role of manure and its importance on the soil system.
* * *
(37) Line 20 page 13: a short conclusion of possible under or overestimation of cloud effect on NO2 VCDs should be useful.

We updated the corresponding section and point out that the shielding effect dominates for cloudy conditions.

**Addition the beginning of section 4.4.3:**
We have investigated possible cloud effects on our results by analyzing the temporal evolution of the mean cloud fraction (CF), $NO_2$ VCDs, and $NO_2$ SCDs around the precipitation event. The latter was added

as it provides the actual measured signal without involving a tropospheric AMF, which is generally very sensitive to clouds.

**Update to the last paragraph of section 4.4.3:**
Interestingly, while there is a strong systematic enhancement of the FRESCO and OMCLDO2 cloud fractions, the $NO_2$ SCDs show no peak around Day0 for GOME-2 and OMI. This indicates that clouds effectively shield the pulsed soil emissions.
* * *
(38) Line 9 page 14: ",thus," may be removed from the sentence.

Done.
* * *
(39) Line 22 page 14: "at" instead of "on" the same latitude.

Done.
* * *
(40) Line 16 page 15: "largest" instead of "larger".

Done.
* * *
(41) Line 3 page 16: specify "Eastern" Sahel, because the analysis has not been made in Western Sahel.

As the global analysis, presented in Fig. 3, indicates enhancements not only in the Eastern Sahel, but also in the western and central part, we would like to keep this sentence.
* * *
(42) Line 5 page 18: As indicated in Aghedo et al., Atmos. Chem. Phys., 7, 1193–1212, 2007, anthropogenic pollution is not likely to reach sahelian latitudes. An important added value could be brought here to this paper. As mentioned in the general comments, the budget over the whole studied area (i.e. Eastern Sahel) could be calculated in TgN for the studied period.

We added the budget calculation (as described at the top of this document) to the end of section 5.1 as well as to the conclusion section. We would like to thank the referee for pointing out that anthropogenic pollution probably does not reach the Sahel. Still, generally higher $NO_2$ VCDs are observed in the Tropics which might be due to biomass burning (anthropogenic influenced) that possibly could reach the Sahel. Therefore, we are quite cautious to exclude this potential interference in the $NO_2$ VCDs over the Sahel region and would like to leave this paragraph as is.
* * *
(43) Conclusions Lines 7-13: difficult to follow with this a) to e) way of presenting the ideas. Proper sentences would be more readable.

We understand the concerns of the referee, but we still think that this way presents our main improvements over previous studies best.
* * *
(44) Line 15 page 18: "maximum amount of precipitation"? do you mean the 2 mm threshold? In that case it is the minimum amount.

The word "maximum" is, indeed, ambiguous in that sense. Referring to the rain threshold "minimum and maximum amount of precipitation" is meant in this sentence. For simplicity, we deleted the word

"maximum" so that the phrase now reads:

(i) evaluate the impact of the a-priori assumptions on thresholds for daily rainfall, i.e. the amount of precipitation and the required duration
* * *
(45) Line 17 page 18: "shown" instead of "showed".

Done.
* * *
(46) Line 20 page 18: again see Oswald et al., 2013, where laboratory measurements made on semi arid soils are presented.

As we only want to indicate our main conclusions of our study we would like to stick to the current text. However, we added the contribution of HONO emissions (Oswald et al., 2013) to the introduction section as described above in the document.

**Technical corrections**
* * *
(47) Line 11 page 3: "lab" should be laboratory

Done.
* * *
(48) Line 13 page 6: Upper case to begin the paragraph is needed.

Done.
* * *
(49) e.g. throughout the text should be in italics.

Done.

---

## Author Comment (AC2) · 3 Jun 2016

**Multi-satellite sensor study on precipitation-induced emission pulses of NOx from soils in semi-arid ecosystems by J. Zörner et al.**

**Reply to anonymous referee #2**

We would like to thank reviewer #2 for the critical and helpful feedback. Our replies as well as the changes made to the manuscript are provided below.

Blue: Reviewer comment
Black: Author's reply
Red: Modified text in manuscript

**General remarks:**

(1) While I have few criticisms of the analysis itself I do have trouble understanding how this analysis significantly enhances our understanding of global soil NOx emissions. Synthesis and interpretation of the results were unsatisfying. For example, there is a confusing amount of time spent on correcting for background emissions following a pulse event. What is the purpose of this? Is it to advance modeling efforts? If so, there needs to be some direct connection of the results to modeling or at least a proposed way in which to use these results to inform modeling.

We think that our study improves the knowledge on precipitation-induced emission pulses of $NO_x$ in many aspects:

- First unambiguous characterization of $sNO_x$ emissions from satellite measurements only with a particular focus on pulsed emissions (no input from models or inventories) and explicit exclusion of other $NO_x$ sources and artefacts
- First global study enabling identification of relatively small-scale areas exhibiting pulsed $sNO_x$ emissions
- Based on the referee comments, a new subsection was added in which total nitrogen emissions from soils are calculated and put into a broader perspective (see comment #1 by referee #1). We thereby set the emissions in context by comparing pulse amounts to regional emissions. This exploits the good time resolution and spatial coverage of the satellite measurements.
- We separated the observed $NO_2$ VCDs into three emission categories: (i) the pulse on Day0, (ii) the enhanced emissions over 14 days following Day0 and (iii) a background which is not directly affected by the pulsing event. This information is beneficial for analogue modelling studies and has not been provided in such detail from a purely top-down approach so far.

We carefully addressed the concerns of the referee in the revised manuscript and also refer to our proposed changes to the manuscript in comment #1 by referee #1.

**Change in the Abstract:**
We find strong peaks of enhanced $NO_2$ Vertical Column Densities (VCDs) induced by the first intense precipitation after prolonged droughts in many semi-arid regions of the world, in particular in the Sahel.

**Addition to line 18, page 18 in the conclusion section:**
Note, however, that our method was optimized for the quantification of pulsed soil emissions from space by demanding long droughts and good viewing conditions (low cloud fractions) on the day of precipitation onset. Thus, regions showing no clear response for these strict selections might still be capable of rain-induced soil emissions.

**Addition to end of the conclusion section:**
With respect to the seasonal $NO_x$ budget, we assess a contribution between 21 to 44% from these rain-induced intense pulsing events to total soil $NO_x$ emissions in the Sahel.
In conclusion, our findings facilitate a detailed characterization and estimation of emission budgets for

intense sNO$_x$ pulses, triggered by individual rain events, which can be directly implemented in modelling studies.
* * *
(2) One of the unique and valuable aspects of the analysis is spatial resolution at the global scale however the authors focus much of the paper on a single event in the Sahel, a phenomenon many other papers have already focused on. The conclusion section does not even mention the global analysis except to say that it was done and confirm that semi-arid regions of the world are likely to have soil NOx pulses. The significance of these pulse emissions at the global scale should be quantified and more clearly presented in order to show their significance within the global NOx budget and how it has advanced our understanding of the global NOx budget.

We first applied our algorithm on a global scale to delineate regions which are suitable for further analyses. The subsequent results indicated that the Sahel region exhibits strongest signals from pulsed soil emissions at the start of the wet season. Therefore, we focused our investigations on this particular region. Our newly introduced budget calculation for the Sahel (see comment #1 of referee #1) shows, that the contribution of the pulsing events to total emissions in the April-May-June period is about 4-8%. Consequently, the overall contribution of such pulsing events to the total nitrogen budget on a global scale is expected to be much smaller. To address the concerns of the referee, we add further conclusions on our global study as well as the budget calculation over the Sahel to section 6. For the corresponding proposed changes to the manuscript, we refer to comment #1, and also to comment #1 by referee #1.
* * *
(3) Also, within the Sahel analysis, it is again not clear how the results enhance our understanding of soil NOx pulses in the Sahel beyond which we already know.

In general, our investigations provide an improved quantification of rain-induced pulsing events of soil NO$_x$ emissions and, furthermore, perform the necessary and so far missing validation work for previous space-based studies in the Sahel. We provide the first unambiguous characterization of pulsed sNO$_x$ emissions from satellite measurements only (no input from models or inventories) and explicit exclusion of other NO$_x$ sources and artefacts. Based on the referee comments, we added a new subsection in which total nitrogen emissions from soils in the Sahel are calculated and put into a broader perspective (see comment #1 by referee #1). Thereby, we separated the observed NO$_2$ VCDs into three emission categories: (i) the pulse on Day0, (ii) the enhanced emissions over 14 days following Day0 and (iii) a background which is not directly affected by the pulsing event. Such detailed information has not been provided by previous studies on pulsed emissions of sNO$_x$ in the Sahel. Furthermore, the high resolution of our approach facilitated the separation of the pulsing signal for different land cover classes in the Sahel region.
We think that the revised manuscript conveys these statements adequately.

**Specific comments**
* * *
(4) On page 3 line 19, it is stated that soil NOx pulses are only enhanced for 1-3 days post precipitation, however other studies have shown pulses to last much longer, up to 25 days. (See Oikawa et al. Unusually high soil nitrogen oxide emissions influence air quality in a high temperature agricultural region, Nature Communications 2015)

We refer in this context to the peak emissions of the pulsing event of sNO$_x$ which typically occur on the scale of 1-3 days (Kim et al. 2012). We updated the corresponding sentence accordingly. Oikawa et al. (2015), however, show that the release of soil NO$_x$ after irrigation peaks approximately seven days after the initial (controlled) wetting and then gradually decreases until the emission rate goes back to background values. This behaviour is different to the one we investigate in our paper and previous space-based studies. As our detailed analysis over specific regions shows, such a delayed pulsing signal cannot be observed using

our approach in the Sahel (Fig. 5), South Africa or Australia (Fig. 13). To address this issue also in our manuscript, we also added a remark to section 5.1.

**Change in the introduction section:**
The main objective of this study is to quantify precipitation-induced short-term enhancements in soil emissions of $NO_x$, which typically show peak emissions on the scale of 1-3 days (Kim et al., 2012), from space-based instruments in semi-arid regions in the world.

**Addition to section 5.1**
Peak emissions of $sNO_x$ pulses typically occur on the scale of 1–3 days (Kim et al., 2012) in accordance to our results for the Sahel, South Africa and Australia showing peak emissions shortly after the first re-wetting. Some studies, on the other hand, measure peak emissions several days after the first re-wetting of the soil, *e.g.* seven days as observed in field by Oikawa et al. (2015). Our algorithm does not specifically distinguish between such cases by taking average time series after the first precipitation event. Single pixels within the regions we investigated may exhibit peak emissions several days after the initial precipitation which would, however, not be resolved by our analysis.
* * *
(5) Pg 6 line 17, Authors state only minor effects resulting from uncertainty in precipitation events across 3 data products. However it would be preferable to quantify that uncertainty or at least state the maximum and average amount of deviation there is across the data products for different regions. Appendix A shows only 1 example.

The average deviations among the three used products (TMPA/TRMM, CMORPH, PERSIANN) are provided for the Sahel, South Africa and Australia as differently grey shaded areas in Fig. 5 and Fig. 10. For a further quantitative assessment of the deviations among the three used products (TMPA/TRMM, CMORPH, PERSIANN) for different regions we refer to literature (Ebert et al., 2007; Novella and Thiaw, 2010; Romilly and Gebremichael, 2011; Liu et al., 2012; Pipunic et al., 2013; Pfeifroth et al. 2016; and references therein).

As appendix A shows, our algorithm does not significantly depend on the choice of precipitation product as *trigger* for the Sahel. Furthermore, we added two maps as appendix E which show enhanced $NO_2$ VCDs for the global study similar to Fig. 3d but for precipitation from CMORPH and PERSIANN products as trigger for the 2 mm threshold. The map based on CMORPH data compares well with our presented TRMM world map in Fig. 3d. The global analysis based on precipitation estimates from the PERSIANN product shows the same spatial patterns, but the general enhancement is lower over the Sahel and higher over South Africa. Although the product choice has some influence on our retrieved signals, this does not affect the observed global spatial patterns. Furthermore, the impact on our budget calculation for the Sahel is also low, as differences are partly reduced by the respective background subtraction and subsequent $sNO_x$ fluxes are of similar magnitude (see appendix E3).

**Change in line 13, page 6, section 2.3:**
Inter-comparison studies show good agreement with ground based precipitation observations for these data products (e.g. Ebert et al., 2007; Novella and Thiaw, 2010; Romilly and Gebremichael, 2011; Liu et al., 2012; Pipunic et al., 2013; Pfeifroth et al. 2016; and references therein) which is, however, variable for different geographic regions, surface types and rain intensities. In our study, we apply each precipitation product individually to differentiate between days with or without rain fall. From the comparison of the corresponding results we find that the uncertainties and differences among the precipitation data sets have only minor effects on the obtained results (see Appendix A, E).

**Appendix E: Impact of a-priori precipitation product**
In this section, the impact of the precipitation product on the derived soil $NO_x$ emissions is investigated. Figures **E1** and **E2** depict the $NO_2$ enhancement on Day0 as in Fig. 3d, but based on CMORPH and PERSIANN data, respectively. While the absolute values differ for the Sahel, the final emission estimates for pulsed emissions are quite similar (see Table **E3**), as the choice of the precipitation data affects the background correction as well.

**E1 CMORPH**

[Figure]

**E2 PERSIANN**

[Figure]

**E3 Derived Fluxes**

|  | **TRMM** | CMORPH | PERSIANN |
|---|---|---|---|
| ∅ Day0 enhanc. [ngNm$^{-2}$s$^{-1}$] | **6.95** | 7.75 | 6.62 |
| max. Day0 enhanc. [ngNm$^{-2}$s$^{-1}$] | **64.61** | 64.61 | 64.61 |
| Day1-14 enhanc. [ngNm$^{-2}$s$^{-1}$] | **3.39** | 3.07 | 2.42 |
| Background (soil) [ngNm$^{-2}$s$^{-1}$] | **2.75** | 3.39 | 4.36 |
| Background (total) [ngNm$^{-2}$s$^{-1}$] | **14.54** | 15.18 | 16.15 |
* * *
(6) Pg.7 line 20. Please provide at least a discussion of the error associated with land cover data sets.

We added remarks on data quality of the GlobCover data set to the paragraph.

The product is publicly available via http://due.esrin.esa.int/page_globcover.php and comprises 22 land cover classes defined with the United Nations (UN) Land Cover Classification System (LCCS) with an overall accuracy across all classes of 58% (Arino et al., 2007; Bontemps et al., 2011). The data is down-scaled using a most-common-value approach to identify dominant land cover types and to match the resolution of the other data sets. Thus, misclassifications might occur particularly over heterogeneous terrain and transition zones, while classification over homogeneous terrain is expected to be robust.
* * *
(7) Line 19 Pg 11–The authors refer to error caused by AMF several times however never indicate any quantification of that error, or suggest references that have investigated error in data products such as in OMI. After filtering for cloud cover, for example, what amount of error is expected to remain?

A general quantification of the error introduced by the AMF is difficult since it is dominated by the uncertainties in assumed trace gas profiles, aerosol and cloud conditions (Boersma et al., 2004). We added the following sentence at the end of section 2.2 to provide a brief remark on the uncertainty for the VCD products:

**Addition to the end of section 2.2:**

Uncertainties of tropospheric $NO_2$ VCDs result mainly from uncertainties of the stratospheric correction (about $2 * 10^{14}\ molec\ cm^{-2}$) and tropospheric AMFs (about 35-60%) (Boersma et al., 2004).
* * *
(8) On pgs 14-15 there is a large discussion of whether enhanced VCD's are the result of precipitation on Day 0 vs precipitation generally being enhanced following that first rain event, aka seasonal changes. For example, the authors state "However, it still needs to be clarified whether the enhanced NO2 VCDs after Day0 are induced by the initial precipitation on Day0 or by continuous precipitation during the following days." But it is not clear to the reader why this distinction is important.

Our study focuses on the quantification of the emission pulse of the first rain of the wet season. Thus, a differentiation between emissions originating from the initial pulse and emissions from other rain events has to be conducted. We also refer to our answer to question #1 of the referee which further describes the need for a separation. We changed the sentence, highlighted by the referee, to:

As our focus is on the quantification of the emission pulse triggered by the first rain of the wet season, it still needs to be clarified whether the enhanced $NO_2$ VCDs after Day0 are induced by the initial precipitation on Day0 or by continuous precipitation during the following days.

---

## Author Response (AR2)

**Multi-satellite sensor study on precipitation-induced emission pulses of NOx from soils in semi-arid ecosystems by J. Zörner et al.**

**Reply to the Editor**

We would like to thank the editor for the helpful feedback. Our replies as well as the changes made to the manuscript are provided below.

Blue: Editor comment
Black: Author's reply
Red: Modified text in manuscript

**General remarks:**
* * *
(1) The claim that the paper presents an approach to quantify soil NOx emissions "globally with a spatial resolution of 0.25 deg" is overstated. I agree that the method can be applied anywhere in the world, but it is only practically useful over selected areas dominated by soil NOx emissions. Much of the paper focuses on the Sahel, as already pointed out by one reviewer. The other two regions discussed in any detail are Australia and southern Africa. Please tone down phrases mentioning the global character of this study, e.g. the "global with spatial resolution of 0.25 deg" aspect of this study in the Abstract and in the Conclusions (page 20, lines 10-11). A more realistic claim would be to state that you have applied the method over all continents, and that it provides meaningful constraints on soil NOx emissions over a number of specific regions that are dominated by soil NOx emissions.

> We agree with the editor that a global quantification of soil emissions is not feasible using our approach. The objective of the algorithm is to derive estimates for pulsed $sNO_x$ emissions. This method was applied globally but is only sensitive to specific semi-arid regions. Thus, we revised the abstract and conclusions to clarify these restrictions.
> **Change at the beginning of the abstract:**
> We present a top-down approach to infer and quantify rain-induced emission pulses of $NO_x$ ($\equiv NO + NO_2$), stemming from biotic emissions of NO from soils, from satellite-borne measurements of $NO_2$.
>
> **Change at the beginning of the conclusion section:**
> We have presented a top-down approach to infer rain-induced emission pulses of $NO_x$ based on space-based measurements of $NO_2$.
>
> **Addition to the abstract and conclusion section:**
> The method is applied globally and provides constraints on pulsed soil emissions of $NO_x$ in regions where the $NO_x$ budget is seasonally dominated by soil emissions.
* * *
(2) The introduction (page 3, paragraph starting on line 10) does not mention at all "canopy reduction", e.g. the phenomenon that nitrogen oxides emitted by soils are quickly deposited on available vegetation surfaces. Still this is an important "environmental condition" influencing the effective soil NOx emission to the atmosphere.

> The removal of $NO_2$ by vegetation was mentioned in the introduction (page 2, line 9). However, we also think this effect should be stressed more with regard to the retrieved soil $NO_x$ fluxes. Thus, we added the role of canopy on the effective soil $NO_x$ to the corresponding paragraph on page 3.
> **Addition to the introduction (line 16, page 3):**
> The effective $NO_x$ fluxes from soils to the atmosphere are potentially offset by "canopy reduction" where nitrogen oxides are quickly deposited on available vegetation surfaces (Ganzeveld et al., 2002).
* * *
(3) In section 2 and later in the paper, I was surprised not to see a discussion on soil temperature data. Controlled experiments and field data suggest that soil NOx emissions are positively correlated with higher soil

temperatures, conducive to stronger microbial activity. This aspect should be discussed.

The dependence of soil $NO_x$ emissions on surface temperature is stated in the current manuscript. This effect, however, was not further discussed in our paper as initial investigations showed that such an analysis entails several difficulties in the interpretation of the results (see the new Appendix F). More detailed analyses would have to be conducted to investigate the sensitivity of the retrieved $NO_2$ pulses to different temperature data sets. This was, however, not within the scope of the current study. By introducing such a temperature separation, the triggered incidences are attributed to different months of the year and locations within the study region (e.g. the Sahel) again complicating the interpretation. It is thereby unclear whether the different $NO_2$ signals are due to specific spatio-temporal characteristics or mainly controlled by the soil temperature. We would therefore like to stay with our initial decision to omit these results from the manuscript, but we will add a short reference to the analysis.

**Addition to the discussion section, page 18, line 17:**

Previous studies also note the strong dependence of soil emissions on temperature. We conducted initial tests using ECMWF soil temperature data (see Appendix F), but found no clear temperature dependence of $sNO_x$ emissions. This is probably due to the systematic relation between temperature and the seasonal cycle which affects the spatio-temporal selection of the data, e.g. in April more southern pixels are selected and in June more northern pixels. In consequence, we indicate the need for more detailed investigation on how these pulsed emissions are affected by different soil temperatures.

**Appendix F: Analysis for different soil temperatures**

In this section, $NO_2$ VCDs are shown (akin to Fig. 5.e) from OMI around the first day of rainfall for different soil temperatures on Day0 for the Sahel region (**left panel**). The **right panel** depicts daily time series of soil temperature (from 12UTC ECMWF data) for the Sahel region (0-30°W, 12-18°N) averaged for the years 2007, 2008, 2009 and 2010.

[Figure]

(4) Another aspect not mentioned but likely playing a role in the analysis, is "smearing" beyond the 0.25 deg cell, or beyond the study area. Suppose grid cell i has its first rain event on day 0, but adjacent grid cell i+1 only has the peak on day 4. In this situation, the time series for grid cell i may be expected to have a local peak on day 0 with an additional enhanced NO2 contribution from day 4 (if downwind of cell i+1). In a general sense, such contributions will influence (raise) the background beyond contributions from sustained local soil NOx emissions alone. This should be acknowledged or further investigated (See specific issues below).

Indeed, such a smearing effect might interfere with our retrieved $NO_2$ time series of the central pixel (which originally experienced the first rain event). This effect, however, also implies that the emitted soil $NO_x$ from pixel i might be advected away (to adjacent pixels) reducing the observed $NO_2$ in pixel i and, thus, under-estimating emissions. In general, the effects of "smearing" were found to be small, mainly because the number of neighbouring pixels (i+n) experiencing their first rain shortly after pixel i is small. The right panel of Fig. 1 basically shows the probability of neighbouring pixels receiving the first rain after pixel i over a course of 60 days. This probability decreases with time and shows no particular drop at 14 days, in contrast to the time series of $NO_2$ VCDs in Fig. 13d,e of the manuscript. Furthermore, the average precipitation in a 10 pixel buffer around pixel i is almost constant over the course of the next weeks (see

left panel of Fig. 1). Thus, we don't think that the enhancement over the two week period following Day0 of pixel i is dominated by this "smearing effect". However, we acknowledge possible artefacts and include a brief discussion on this in sections 4.5 and 5.1.

[Figure]

Figure 1: **Left:** Average precipitation over 60 days following Day0 in pixel i (red) and for the neighbouring 10 pixel buffer (blue). **Right:** Number of pixels in a 10 pixel buffer around pixel i [in %] experiencing the first rain event itself.

**Addition to section 4.5, page 15, line 10:**
This could be related to inflow of soil $NO_x$ from adjacent pixels which receive the first rain shortly after. However, initial tests showed that the number of such incidents is quite small (less than 5%). For this reason, and because the effect is possibly offset by sNOx advection out of the pixel of interest, it is assumed that the effect of inflow is not dominant. As we focus on events with strong emissions (compared to background), a "smearing effect" by advection leads to a consistent underestimation of the peak and subsequent emissions. Thus, we conclude that inflow cannot explain the enhancement for 14 days after the first rain event. Furthermore, it remains unclear to what extent the enhanced $NO_2$ VCDs are affected by a possible underlying change in the background.

**Addition to section 5.1, page 18, line 18:**
Possible advection effects into or out of the considered pixels may thereby raise or lower the retrieved $sNO_x$ fluxes. The analysis based on different dry phases after Day0 also changes the probability for first rain events in pixels in close proximity. As we do not find strong differences in the emissions for different dry phases after Day0, we conclude that the inflow of $NO_x$ from adjacent pixels is not the dominant source for the enhancement after Day0. In contrast, a systematic underestimation of the emissions is likely due to advection out of the central pixel. In sum, we estimate that the integrated emissions after Day0 are potentially of the same order of magnitude as those from the first emission pulse on Day0.
* * *
(5) The suggestion to implement your findings in models is a good one. I strongly encourage you to provide modellers with some practical hints as to how to do this. Preferably you give an equation that represents a parameterization of the pulsing events. Such an equation takes into account the most important drivers of the pulsing events: length of dry spell, amount of precipitation of Day0, land cover type. It is important to also quantitatively relate the pulsing emissions to the non-pulsing soil NOx emissions. Models generally include parameterizations for those (Steinkamp and Lawrence, Hudman-papers), and these would have to be tweaked if a pulsing contribution is accounted for. If you could show how this could be done, this would add a lot of value to your paper.

To our knowledge, our study provides the first clear separation of the different emission types: (i) pulsed emissions on short time scales, (ii) enhanced emissions after the initial pulse, and (iii) seasonal background emissions. We see this separation as a first step for an improved parametrization of soil emissions in models. However, for a quantitative parametrization based on our approach all tests for different land

covers, drought lengths, precipitation on Day0 and possibly temperature ranges need to be run consistently with the corresponding background corrections. Thus, at the moment we advise modellers to treat pulsed emissions of soil $NO_x$ not as singular events but also as impact on the background thereafter. We updated the last sentence of the conclusion to account for the current limitation of a possible model implementation: In conclusion, our findings facilitate a detailed characterization and estimation of emission budgets for intense $sNO_x$ pulses, triggered by individual rain events, which can improve parametrizations in modelling studies by dividing soil emissions into several parts: (i) pulsed emissions on short time scales, (ii) enhanced emissions after the initial pulse, and (iii) seasonal background emissions.

**Minor issues:**
* * *
(6) P2, l20: please also include by Vinken et al., ACP, 2014 as a study indicating that soil NOx emissions are regionally underestimated.

 Done.
* * *
(7) P2, L27-29: suggest to merge the sentence: "Findings ... from soils" with the previous paragraph.

 Done.
* * *
(8) P3, L21: also the study from Bertram et al., GRL, 2006 should be mentioned here.

 Done.
* * *
(9) P5, L2-4: the AMF also depends on the satellite viewing geometry (solar and viewing zenith angles, relative azimuth angle).

 We updated the sentence accordingly:
 The AMF is derived from radiative transfer simulations taking into account information of ground albedo, aerosols and clouds, the vertical profile of the trace gas and the satellite viewing geometry (e.g. Palmer et al., 2001; Richter and Burrows, 2002; Martin, 2003).
* * *
(10) P5, L13-15: please check carefully with the documentation on www.temis.nl or with Ronald van der A whether FRESCO or FRESCO+ was used in your GOME-2 and SCIAMACHY product versions. If FRESCO+, please include the appropriate citation.

 We updated the paragraph and references (also in the abbreviation table) now referring to FRESCO+.
* * *
(11) P5, 21: swath should be indicated as 2600 km, not km2.

 Done.
* * *
(12) P5, L28-29: the instantaneous NOx lifetime and emission diurnal cycle also contribute to the differences between SCIAMACHY, GOME-2 on one hand, and OMI on the other. This should be acknowledged. Generally NO2 in the early afternoon is shorter-lived than in the morning. This effect may (partly) offset by soil NOx emissions peaking in the afternoon rather than in the morning (higher temperatures).

 This is a valuable aspect which is now added to the corresponding paragraph:
 **Addition to page 5, line 30:**
 Furthermore, the diurnal cycle of the instantaneous $NO_x$ lifetime and emissions might also cause systematic

differences between SCIAMACHY and GOME-2 on the one hand, and OMI on the other.
* * *
(13) P10, L15: 50% of possible data available per pixel because of reasons of spatial representativeness? For a detailed discussion of this aspect, please see Boersma et al., GMD, 2016. That paper discusses how uncertainty from spatial representativeness becomes the dominant uncertainty contribution if only few valid satellite pixels per grid cell are available. It would be worthwhile (for future proper use of the satellite data) to discuss the issue briefly and refer to the Boersma et al. [2016] paper.

We updated the sentence to explicitly regard the spatial representativeness aspect:

Furthermore, as the uncertainty from spatial representativeness becomes the dominant uncertainty contribution if only few valid satellite pixels per grid cell are available (Boersma et al., 2016), at least 50% of possible data are requested per grid pixel in order to be considered in this analysis.
* * *
(14) P14, L14-15: the pixel size differences between GOME, SCIAMACHY, and OMI also lead to different CF distributions.

We added this remark to the corresponding sentence:

The different temporal variation might also be partly related to the different overpass times and pixel sizes among the three satellite instruments.
* * *
(15) P14, L21: typo Fig. 9.d.

Corrected.
* * *
(16) P14, L25L typo "are" should be 'is' (the influence is investigated).

Corrected.
* * *
(17) P15, L12-16: in this section also smearing should be discussed. To me it seems unlikely that over a larger region like the Sahel, all your pixels have the first rain on the same Day0. If the first days of rainfall vary over a region, then the background for pixel i will certainly be influenced by advection of soil NOx peaking earlier or later than Day0. This should be acknowledged and preferably also be inestigated in more detail (e.g. by selecting only those pixels from the Sahel that have the exact same Day0. If the distribution of Fig 12(a) does not change appreciably for this subset, then smearing may be of little importance.
P16, L30-32: "NO2 VCDs stay enhanced after Day0 for a period of about two weeks". In line with my previous comment, this may be a convolution of structurally enhanced soil NOx emissions (microbes activated by the first rains) and the result of smearing. Your paper would be more convincing if you could wiggle out the effect of smearing.

For our elaborated response please see comment #4.
* * *
(18) P17, L24: "with an approach similar to ours". This sounds strange, since Hudman's approach was published well before this paper. I suggest to revert the statement and keep it at that your approach is similar to theirs.

Since this remark has already been stated in the introduction, we updated the sentence as follows:

The latter study investigates pulsed soil emissions of $NO_x$ in the Sahel region and finds a comparable magnitude (49% relative increase of OMI $NO_2$ VCDs on Day0) and length of the pulsing event (1-2 days) following the first rainfall.
* * *
(19) P19, L7-12: I had to read a couple of times before I understood that you arrive at the 21-44% estimate for the contribution of pulsing to total soil NOx emissions (10-21 GgN) by dividing your own pulsing estimates by the enhancement of the background (46.4). Please clarify this in 5.2.

We rephrased the paragraph to:

Scaling up the Day0 emissions results in 1.2 GgN and 12 GgN, considering the lower and upper flux estimates estimated above. Analogously, the emissions over the following two week period add up to 8.8 GgN. Together this sums up to 10.1-20.8 GgN emissions due to pulsing. As mentioned above, the observed increase of the background in the AMJ-period of $0.17 * 10^{15}$ molec cm$^{-2}$ is mainly driven by microbial emissions from soils as well. When integrated over the complete April-May-June period, these seasonal soil emissions correspond to 46.4 GgN (again based on a NO$_x$ lifetime of 4 hours). Consequently, the emissions due to pulsing contribute about 21-44% additionally to seasonal soil emissions for the Sahel and dominate the local NO$_x$ concentrations on the particular days.

—————————

(20) P20, L12: Hudman should not be between brackets.

Done.

—————————

(21) P20, L34: "NO2 VCDs show moderately enhanced NO2 VCDs" ... this should be rephrased.

We rephrased the sentence in the abstract and the conclusions as follows:

[revised manuscript text omitted]